# Gene Characterization of Nocturnin Paralogues in Goldfish: Full Coding Sequences, Structure, Phylogeny and Tissue Expression

**DOI:** 10.3390/ijms25010054

**Published:** 2023-12-19

**Authors:** Diego Madera, Aitana Alonso-Gómez, María Jesús Delgado, Ana Isabel Valenciano, Ángel Luis Alonso-Gómez

**Affiliations:** Departamento de Genética, Fisiología y Microbiología, Universidad Complutense de Madrid, 28040 Madrid, Spain; dmadera@ucm.es (D.M.); aitaalon@ucm.es (A.A.-G.); mjdelgad@ucm.es (M.J.D.); aivalenc@ucm.es (A.I.V.)

**Keywords:** nocturnin nomenclature, *Carassius auratus*, zebrafish, molecular cloning, gene evolution, synteny analysis, synteny quantification, phylogenetic tree, protein structure, tissue distribution

## Abstract

The aim of this work is the full characterization of all the nocturnin (*noc*) paralogues expressed in a teleost, the goldfish. An in silico analysis of the evolutive origin of *noc* in Osteichthyes is performed, including the splicing variants and new paralogues appearing after teleostean 3R genomic duplication and the cyprinine 4Rc. After sequencing the full-length mRNA of goldfish, we obtained two isoforms for *noc-a* (*noc-aa* and *noc-ab*) with two splice variants (I and II), and only one for *noc-b* (*noc-bb*) with two transcripts (II and III). Using the splicing variant II, the prediction of the secondary and tertiary structures renders a well-conserved 3D distribution of four α-helices and nine β-sheets in the three *noc* isoforms. A synteny analysis based on the localization of *noc* genes in the patrilineal or matrilineal subgenomes and a phylogenetic tree of protein sequences were accomplished to stablish a classification and a long-lasting nomenclature of *noc* in goldfish, and valid to be extrapolated to allotetraploid Cyprininae. Finally, both goldfish and zebrafish showed a broad tissue expression of all the *noc* paralogues. Moreover, the enriched expression of specific paralogues in some tissues argues in favour of neo- or subfunctionalization.

## 1. Introduction

Nocturnin (NOC) is an enzyme discovered in the retinal photoreceptors of *Xenopus laevis,* where it displays a daily rhythm of expression peaking at the beginning of the scotophase [1]. Day–night variations in the levels of *nocturnin* (*noc*) gene expression have been described in species belonging to different animal phyla. In Porifera, the *noc* expression is found in the demosponge *Suberites domuncula* in the epithelial layer around the aquiferous canals, showing maximal levels in darkness [2]. In insects, a daily rhythm of *noc* expression is described in the brain of female mosquitos [3], and in both the head and trunk of the fruit fly (*Drosophila melanogaster*) [4]. In vertebrates, a ubiquitous pattern of *noc* expression is also present from fish [5,6] to mammals [7,8], including the central nervous system (retina and encephalic areas) and most of the organs. In amphibians, *noc* was detected by in situ hybridization in most of the tissues during the early development of clawed frog (*Xenopus laevis)* [9]. In mammals, a daily variation of mouse *noc* expression has been also reported in the retina and brain areas, and with higher amplitude in non-neural tissues [7]. Besides, a daily variation of *noc* expression has also been detected in human peripheral blood lymphocytes [10].

The nuclear regulation of *noc* expression has been only studied in a few species of vertebrates. In *X. laevis* photoreceptors, the binding of pCREB (phosphorylated cAMP response element-binding protein) to a sequence in the nocturnin promoter activates the transcription of *noc*. Light inhibits CREB phosphorylation, while darkness induces pCREB and, therefore, the transcription of *noc* gene [11]. In a hepatoma cell line, *noc* expression is controlled by the binding of the CLOCK/BMAL1 heterodimer to an E-box element of the *noc* promoter, similarly to the activation of other clock-controlled genes [8].

Possible functions of NOC remain quite elusive to date. Due to its structural homology with the ccr4 (carbon catabolite repression 4) family, NOC was first linked to the regulation of the mRNA half-life in cells, removing the poly-A tail of some clock genes and clock-controlled genes [12,13], and therefore with a possible role in the circadian system [14]. Further studies demonstrated that the main enzymatic activity of NOC is the dephosphorylation of NADP(H) to NAD(H) [15], linking NOC with the metabolic homeostasis as a switch between catabolism and anabolism. Accordingly, some data relate *noc* expression with the metabolic control of energy balance, and this possible relationship seems to be well-conserved throughout phylogeny. Thus, in demosponges, the daily rhythm of *noc* expression is inverted with respect to glycogenin [2]. In fruit flies, *noc* expression is upregulated after starvation [4]. In fish, feeding regulates *noc* expression in the hypothalamus, hepatopancreas, and intestinal bulb [6]. In mammals, knockout mice (*noc*−/−) showed resistance to obesity and hepatic steatosis when they were fed a high-fat diet [16]. Furthermore, the *noc*−/− mice have also deficits in the intestinal triglyceride absorption [17] and dysregulation of lipids and glucose metabolism [18,19].

In addition to metabolic functions, NOC is also involved in other functions; for example, it may act as a non-enzymatic transcription co-factor by promoting the transcription factor PPAR-γ to translocate into the nucleus of fat cells [20], enhancing the differentiation of preadipocytes to adipocytes. Moreover, NOC negatively regulates osteogenesis by inhibiting bone IGF-1 expression [14,21], and *noc* is critical for the development of early mouse embryo [22], in the timing and differentiation of somites in amphibians [23], and in wing morphogenesis at the pupal stage in fruit flies [4].

A full-length cDNA sequence of *noc* has been cloned to date only from the retina of the clawed frog [1], mouse [7], and one of the two *noc* paralogues (*noc*-*b*) in zebrafish [5]. Besides, partial cDNAs from humans, cows, chickens, zebrafish, and goldfish have also been obtained [5,6,7]. Nowadays, thanks to massive sequencing technology, the entire genomes of many vertebrates are now available. The in silico analysis of these data allows us to conclude that all vertebrates so far studied have at least one gene coding for *noc*, with the same basic structure consisting in three exons with a highly conserved sequence.

Teleosts are a very interesting vertebrate group used to address genomic studies, because of the whole-genome duplication (named 3R) undergone by their ancestor [24]. This fact determines that current teleosts should have at least two gene paralogues for *noc*. In gene sequence databases, these two genes were firstly called “*nocturnin*” and “*nocturnin-like*”, respectively, being later renamed *nocturnin-a* and *-b* (*noc-a* and *noc-b*), based on full and partial sequences obtained from cDNA in zebrafish [5] and goldfish [6]. Particularly, both salmonids [25] and the subfamily Cyprininae (to which goldfish belongs) [26] go through an additional whole-genome duplication, known as 4Rs and 4Rc, respectively. Thus, the presence of four *noc* paralogues in the genome of goldfish is expected.

The aim of this work is to carry out the first full characterization of all *noc* paralogues expressed in the Cyprininae subfamily. We have obtained the complete coding sequences of all splicing variants of *noc* paralogues expressed in the goldfish transcriptome, as well as the mRNA non-coding regions (5′UTR and 3′UTR). A comprehensive in silico analysis of the evolutionary profile of these genes has been performed, including the synteny, phylogenetic tree, and prediction of secondary and tertiary protein structures deduced from translated amino acid sequences. Furthermore, we have studied the tissue expression of *noc* transcripts in two species of cyprinids, goldfish (*Carassius auratus*) and zebrafish (*Danio rerio*). This full characterization of all the *noc* paralogues in the Cyprininae subfamily lays the essential groundwork for further functional studies of this intriguing enzyme.

## 2. Results

### 2.1. Teleostean Nocturnins: Evolutive Origin

The general structure of the gene encoding *noc* has not changed throughout the teleost phylogeny (Figure 1). Taking the spotted gar (*Lepisosteus oculatus*), a non-teleost model as ancestral Osteichthyes prior to the 3R duplication, we found that the single *noc* gene has three exons and two introns (Gene ID: 102682701). A detailed search in the genomic sequences of this species allows us to propose the existence of three alternative sequences for the first transcribed exon. We named these exon 1a, exon 1b (non-coding), and exon 1c. By means of alternative splicing, three different transcripts for *noc* may be generated. Based on the position of transcription start sites (TSS) in the gene sequence, we named them variant I (with exon 1a, accession no. XM_006629623.2), variant II (with exon 1b, accession no. XM_015345017.1), and variant III (with exon 1c, accession no. XM_015345016.1). To date, no experimental confirmation of these variants on transcriptomes has been published.

The 3R genomic duplication in the common ancestor of teleosts (350 million years ago) resulted in two forms of *noc* (*noc-a* and *noc-b*) that were initially identical but have accumulated several divergences (by sub-functionalization and/or neo-functionalization) during their subsequent evolution, which allowed their conservation in current teleosts. The analysis of the sequences shows that *noc-a* seems to retain the three alternatives for exon 1 in many species, but the *noc-b* gene loses the exon 1a (splice variant I) in the ancestral Clupeocephala (advanced teleosts: Otomorpha plus Euteleostei), since the basal teleostean fish *Anguilla* sp. (accession no. XM_035419224.1) and *Scleropages* sp. (accession no. XM_018737923.2) do conserve exon 1a. Then, *noc-b* for Clupeocephala only has two splicing alternatives (forms II and III, Figure 1).

Furthermore, the *noc-b* gene structure underwent additional changes in teleostean fish, a new short intron (phase 0) splits exon 3 into two parts in the Euteleostei clade (e.g., *Salmo salar*, Gene ID: 106610553), and other phase-0 intron splits exon 2 in the Neoteleostei clade (e.g., clown anemonefish, *Amphiprion ocellaris* Gene ID: 111585149); however, these insertions did not break the coding sequence of *noc-b*. In the Ostariophysi clade, these new intron insertions were not observed, preserving the ancestral structure. In orders Characiformes, Siluriformes, and Gymnotiformes, the *noc-b* gene has been entirely lost.

The four paralogues for *noc* that would be expected after the additional duplication 4Rc of the Cyprininae subfamily are confirmed in genomic databases of *Cyprinus* sp. and *Sinocyclocheilus* sp. The 4Rc duplication generates allotetraploids, where the karyotype is divided into two subgenomes with a female or male origin. Then, we named these *noc-aa*, *noc-ab*, *noc-ba*, and *noc-bb*, where the last letter indicates patrilineal (a) or matrilineal (b) subgenomes. The cyprinine *noc-a* paralogues (*noc-aa* and *noc-ab*) have been annotated in databases only with two splicing variants, I and II, suggesting the loss of variant III (Figure 1).

After sequencing the *noc* mRNA of goldfish (Figure 2), we obtained evidence of the two isoforms for *noc-a* (*noc-aa* and *noc-ab*), with only two splice variants (I and II). For *noc-b*, only one paralogue (*noc-bb*) has been found with two possible transcripts (II and III). Despite not having found any trace of *noc-ba* in our goldfish mRNA samples, it is possible that *noc-ba* could exist in the goldfish genome, as it is found in other members of the Cyprininae subfamily that also underwent the 4Rc duplication. Therefore, we extend our search to genome projects.

Accessing the complete genome of *Carassius gibelio* (Prussian carp) and *Carassius auratus*, we found the paralogue *noc-ba* with a high identity with the *noc-ba* sequences of other cyprinines (such as different species of *Sinocyclocheilus* and *Cyprinus carpio*). However, the *Carassius* sp. sequences show several mutations, especially large deletions in the coding part, resulting in truncated and non-functional coding sequences (Figure 3). Some of these deletions are identical in *C. auratus* and *C. gibelio*. Such deletions are confirmed to be real, and are not artifacts from the sequence assembly process. The consequence of frameshift deletions is the appearance of premature stop codons in exon 3 (13 red stars in *C. gibelio*, and 3 in *C. auratus*, Figure 3).

### 2.2. Nocturnin Sequences in Goldfish

#### 2.2.1. Transcription of *Nocturnin aa* (*noc-aa*) and *Nocturnin ab* (*noc-ab*) Genes

Sequencing of mRNA allows for the description of complete coding sequences for *noc-aa* splicing variants I and II (Figure 4). As expected from proposed cyprinine *noc* evolution (Figure 1), no trace of variant III was found. Exons 2 and 3 are common to both variants (I and II); however, the exon 1 is different. For variant I, there is a long coding sequence (202 bp) from initial methionine (exon 1a). Variant II has a different exon 1b, with no available methionine codon (ATG). Instead, the first codon ATG in exon 2 initiates variant II (green box in Figure 4).

By cloning the RT-PCR amplicons, we resolved the multiple single-nucleotide polymorphisms (SNPs) that we found using conventional PCR. We have identified two alleles of the *noc-aa* paralogue, with two splicing variants I and II, respectively (Figure 4). The differences between both allele sequences are in the 5´UTR and in the exons 1 and 2, showing 11 changes in nucleotides. Despite this, we found only four amino acid conservative changes (Val/Leu^82^, Thr/Ala^89^, Glu/Asp^97^, Tyr/His^101^) that did not affect the structure of protein and were located far from the catalytic site.

As described for *noc-aa,* two transcripts are sequenced for the *noc-ab* gene (Figure 5), corresponding to the predicted splicing variants I and II. Variant II is not annotated in any available *Carassius* genome (NCBI and ENSEMBL databases). The short length of this non-coding sequence (exon 1b) may hinder detection during the automatized annotation process. However, variant II was recovered by our experimental mRNA sequencing or by searching in transcriptomic libraries (Appendix A). In this case, we did not detect alleles for *noc-ab*.

We determined that both *noc-aa* and *noc-ab* paralogues have a very similar structure with many common features (Figure 4 and Figure 5). The stop codon TAG terminates the coding sequence, insertions of introns are in phase 1, and the separation between exons is identical, being a GTG codon in the transition of exon 1–exon 2 and a GCT codon in the case of exon 2–exon 3. The initial methionines of the splicing variants I and II have the same relative position in the coding exon 1a for variant I (in cyan) and exon 2 for variant II (in green). We correctly verified the position of initial methionines, because of the presence of many in-frame stop codons upstream (bold letters in 5’UTRs, Figure 4 and Figure 5).

#### 2.2.2. Transcription of *Nocturnin bb* (*noc-bb*) Gene

We have identified two splicing variants (II and III) for the *noc-bb* paralogue (Figure 6), with the two corresponding starting ATG codons. Exons 2 and 3 are common to both variants (II and III), but the presence of a short coding exon 1c (13 pb, pink box in Figure 6) is specific to variant III. The corresponding sequence (MEVVA-) fulfils the requirement of the DEG Nend UBRbox 2 motif (Appendix A).

The separation between exons is different compared to both *noc-a* paralogues, with a GCA codon in the two links between exons. In addition, the sequence of the stop codon is TAA, showing another difference from *noc-a* paralogues. For the *noc-bb* transcript, we reached the poly-A tail (not included in Figure 6).

#### 2.2.3. Protein Alignment and Deduced Secondary Structure

Variants I of *noc-aa* and *noc-ab* are characterized by a well-conserved long first coding exon (60–65 amino acids), described as a mitochondrial-targeting signal (MTS) in mammals. We performed the analysis of variant I of cyprinid nocturnins, looking for MTS motifs by using MitoFates and TPpred 3.0 tools. TPpred is more sensitive than MitoFates for detecting MTS, and our results confirm that the variant I contains a real MTS (Appendix A), while variants II and III lack any trace of the MTS motif.

Alignment of the amino acid sequences deduced from the three paralogues of goldfish nocturnins with the matching fragments of the sequences from other vertebrates was performed (Figure 7). The splicing variant II was used for the three isoforms because it is the core sequence common to all vertebrates. The result reveals a high conservation of these sequences, especially in the central zone and the C-terminal region. In the alignment, we identified amino acids linked to the catalytic activity of the human NOC, verifying its conservation throughout the evolution, represented by green boxes in the alignment (Figure 7: Asn^149^, Glu^195^, Lys^219^, His^286^, Lys^288^, Arg^290^, Asp^324^, Asn^326^, Arg^367^, Asp^377^ and His^414^). The leucine zipper motif is a segment of 30 amino acids organized in an α-helix, with periodic repetition of leucines in every seventh position and their side chains line up on the same side of the α-helix. In the alignment, a region which resembles this structure can be observed, with some variations. The fourth leucine residue has mutated to threonine in cyprinids, the third leucine in goldfish NOC-AA is replaced by a methionine (a hydrophobic amino acid like leucine), and in the case of humans, the third leucine is replaced by a phenylalanine (also hydrophobic). Only *Xenopus tropicalis* retains this motif unchanged.

Other conserved features of NOC proteins may be related to their function or subcellular location, as indicated by the presence of an N-myristoylation motif (MGXXXS) in the N-terminal position (red rectangle in Figure 7) and the nearby Ser/Thr-rich region (orange rectangle in Figure 7). All cypriniform NOCs (variant II) have a high confidence score from the Myristoylator program, higher than 0.91, except *Carassius gibelio* NOC-AB, *Cyprinus carpio* NOC-AB, and NOC-BA, in which the N-myristoylation motif was lost (Appendix A).

A prediction of the secondary structure of the three NOC isoforms obtained with JPred4 is shown in Figure 7. Those results with values of low confidence and with less than three residues for either α-helix or a β-sheet are not indicated in the figure. We observed a total of four α-helices and nine β-sheets conserved in the three *noc* isoforms of goldfish.

#### 2.2.4. Tertiary 3D Structure 

The prediction of the tertiary structure of NOC proteins was obtained with the SWISS-MODEL program, which creates a possible tertiary structure from an aminoacidic sequence, taking an evolutionary-related protein as template. The crystallized structure of the human NOC catalytic domain has been used as a model (PDB accession number: 6bt1) for the prediction of the tertiary structure of the three splicing variant II of goldfish nocturnins. The human template was truncated (without the 119 N-terminal amino acids containing the MTS motif) to obtain the correct crystallization of the core domain. The results show a well-conserved 3D distribution of α-helix and β-sheet motifs (blue domains, Figure 8) in goldfish. Meanwhile, only the small peripheral loop (orange) does not preserve its spatial structure. These non-conserved regions, between α4 and β9, match with the sequences inside the black box of the protein alignment (Figure 7).

### 2.3. Proposed Classification for Cyprinine Nocturnins

Our next objective was to establish a classification and a long-standing nomenclature of nocturnins throughout the teleost phylogeny. We took the zebrafish as a base, where a clear distinction between *noc-a* and *noc-b* exists, corresponding to specific teleost WGD or 3R. We performed a complete synteny analysis and created a phylogenetic tree of protein sequences.

#### 2.3.1. Synteny Analysis

Synteny analysis allows us to determine the conservation of the closest loci of *noc* in goldfish and four Cyprinidae species: *Danio rerio* (zebrafish), *Sinocyclocheilus grahami* (golden-line barbel), *Cyprinus carpio* (common carp), and *Carassius gibelio* (Prussian carp). For this task, we used the publicly accessible database indicated in Appendix A. In addition, we used the comparative analysis of these five species to identify loci that can be used to discriminate between matrilineal and patrilineal chromosomes, and that could be used as predictors in other Cyprininae species.

Figure 9 shows the synteny analysis of *noc-a* paralogues. There is a strict conservation of genes upstream and downstream of *noc-a*, from *ccdc88b* to *naa15a*, in all orthologous chromosomes. The paralogue *elf2a*, downstream neighbor of *noc-a*, may be used as a good predictor of *noc-a* paralogues’ location in other cyprinid species. The *flrt1b* gene is inserted into an intron of the *macrod1* gene in the opposite direction of transcription. In Figure 9, *flrt1b* appears as a gap at the center of the *macrod1* gene.

The search for discriminant (matrilineal–patrilineal) loci allowed us to identify the *fgfbp1b* gene downstream of *noc-aa*, and an uncharacterized gene between *scyl1* and *ccg8* upstream of *noc-aa*, which are specific and diagnostic patrilineal markers. A duplication of *rarres3* in *Carassius gibelio* is conserved in *C. auratus*. The respective matrilineal chromosome of cyprinines is characterized by the loss of several genes, such as the *fgfbp1b* gene located between *anxa5a* and *fgfbp2a* in zebrafish. Furthermore, there is a point deletion of *ugl*, placed between *naa15a* and *ra33ba*, and the *btbd18* deletion located between *cryba1l1* and *tmem33*; both deletions are shared by *C. gibelio* and *C. auratus*.

Figure 10 shows the synteny analysis of *noc-b*. The gene *elf2b* is located downstream of *noc-b* in all the orthologous chromosomes. Upstream of *noc-b*, chromosomes show a large region with no genes annotated up to *pcdh10a*. Chromosomes in Figure 10 are not to scale, and this gap is not represented. The chromosome of the golden-line barbel is represented in two separate segments, because the assembly of its genome is incomplete.

The map shows the presence of *wfs1a* and *slc34a2a* downstream from *noc-bb*, and an uncharacterized gene upstream from *noc-bb* as adequate markers of matrilineal origin of these chromosomes. In *Carassius gibelio*, it is important to consider a deletion of a cluster between *elf2b* and *pde5aa*. However, in goldfish, the insertion of *pgbd3* gene and an inversion of loci from *tcr-like* to *ppp2r2ca* is more representative. Regarding the patrilineal chromosomes, wherein *noc-ba* resides, we have determined the presence of two zinc finger genes upstream of *noc-ba*, with *zmym1* being conserved in common carp, Prussian carp, and goldfish, and *zbed4* in species of *Carassius*. The absence of this segment in the golden-line barbel genome impedes their accurate identification. In all the cyprinines analyzed, pseudogenization may be a frequent process in these patrilineal chromosomes, and we can use it as a diagnostic tool. For example, *pde5aa* in all species and *stim2a* and *noc-ba* in *Carassius* species are pseudogenized with the accumulation of several deleterious mutations. In addition, the *drd1* gene has tandem-duplicated in goldfish and common carp with a pseudogenized copy. Prussian carp (*C. gibelio)* has lost completely this pseudogene.

To quantify the similarity of two syntenies, we proposed two quantitative parameters: the synteny index (SI, related to the number of relative positions conserved) and the synteny conservation rate (SCR, as a percentage of maximum theoretical synteny conservation). These numerically indicate the conservation of relative gene positions in two species. The detailed definitions and calculation methods of both parameters are in Appendix B.

We compare goldfish with Prussian carp, common carp, and zebrafish, according to its decreasing phylogenetic distance (Figure 11C). We exclude golden-line barbel due to the fragmentary status of its genome project. In the orthologous comparison (matrilineal to matrilineal, or patrilineal to patrilineal), the SI is higher than 1 (Figure 11A) and the SCR is over the 50% (Figure 11B) for all *noc* paralogues, indicating that the closest loci to *noc* are very similar in Cyprinidae. As expected, when goldfish and zebrafish were compared, both parameters show low values, in agreement with their larger phylogenetic distance (different subfamilies). The comparison zebrafish *noc-b* vs. goldfish *noc-ba* has an exceptionally low synteny conservation (SI: 1.4 and SCR: 59%), because of many mutations (inversions and indels of individual genes) on goldfish chromosome 1. They have a more marked effect in reducing synteny indexes than the single large inversion on chromosome 26.

In Cyprininae, *noc-ab* and *noc-ba* synteny conservation follows the expected genetic distance, i.e., higher values for Prussian carp vs. goldfish than the common carp vs. goldfish comparison. Furthermore, *noc-ab* reached the maximum synteny conservation rate (mSCR, Appendix B) of 94% in both *Carassius* species, indicating the perfect synteny of these chromosomal segments (Figure 9). In *noc-aa* and *noc-bb*, we found that the synteny conservation is higher for common carp vs. goldfish than for Prussian carp vs. goldfish. This paradoxical result may be explained by the evident changes in Prussian carp chromosomes, as the multiple loci deletion in chromosome B1, and the inversion of *scyl1*-*hrasls3* segment in chromosome A14.

The high values of these two numerical parameters are only obtained from the orthologous contrast test, meaning comparison of matrilineal–matrilineal or patrilineal–patrilineal chromosomes (Figure 11). In the ohnologous contrast test (comparison of matrilineal to patrilineal and vice versa), the synteny index drops by two to ten times, and synteny conservation rate is reduced from a half to a third (Appendix A).

#### 2.3.2. Phylogenetic Tree

A phylogenetic tree was designed with the amino acid sequences of the NOC splicing variant II of each paralogue, except for *H. sapiens*, *Sinocyclocheilus rhinocerous*-BA, *S. anshuiensis*-AA and AB, *S. grahami*-AA, and *Cyprinus carpio*-AB, where the N-truncated form of NOC was used to eliminate exon 1a of splice variant I, when variant II was not available in GenBank. Figure 12 shows the result obtained by the ML method. The NOC from Sarcopterygii (*H. sapiens*, *X. tropicalis*, *Latimeria chalumnae*) and the ancestral Osteichthyes (*Lepisosteus oculatus*) appears as a single gene in the basal position of the tree. Teleosts nocturnins are divided into two main clades (NOC-A and NOC-B) with a mean of 60% identity. This gene duplication corresponds to teleostean 3R. As expected, in the most complete teleostean genomes available, NOC-A and NOC-B always resides in different chromosomes. *Lepisosteus oculatus* has a single *noc* gene, which has not undergone the 3R duplication. To assess the nomenclature A or B, we follow the proposed for zebrafish nocturnins.

The tree reveals the main division of Clupeocephala (Otomorpha and Euteleostei) as monophyletic branches for both NOC-A and NOC-B. On the other hand, the phylogenetic analysis makes it possible to locate the full genome duplication of the Cyprininae subfamily (4Rc) in the tree, with moderate bootstrap support (above 667). From this node, we can observe a symmetrical bifurcation point with representatives in each branch of all cyprinine species analysed. We choose the branch nomenclature AA, AB, BA, and BB, to show the allotetraploid origin of cyprinines, and the patrilineal (genome A) or matrilineal (genome B) subgenomes (Figure 9 and Figure 10). The close relationship of the NOCs inside each branch (identity of about 90% for AA-AB branch and 88% for BA-BB branch), in all cyprinines (*Sinocyclocheilus* sp., *Cyprinus* sp., *Carassius* sp.) allows us to verify that the names assigned to the *noc* are correct. The goldfish NOC-BA paralogue is connected to the NOC-BA cluster with a long branch caused by the high number of mutations in this pseudogene. However, NOC-BA from *Cyprinus* and *Sinocyclocheilus* are functional genes and are located in the expected position in the tree, with a similar branch length compared to other nocturnin paralogues.

### 2.4. Tissue Expression of noc mRNA in Goldfish and Zebrafish

The relative gene expression of *noc-aa*, *noc-ab* and *noc-bb* in goldfish is shown in Figure 13. The three paralogues have a ubiquitous expression, with a distinctive pattern for each paralogue. In peripheral tissues, the adipose tissue showed the highest expression of the three noc paralogues, with significant lower values in the spleen, hepatopancreas, skin and muscle (Figure 13A,C,E). The lowest expression of these paralogues was found in the gastrointestinal tract.

In nervous system, both *noc-aa* (Figure 13B), and *noc-ab* (Figure 13D) are highly expressed in the pituitary in contrast to the other brain areas. Moreover, *noc-aa* (Figure 13B) transcripts are twice as abundant in the diencephalon and cerebellum with respect to the telencephalon, optic tectum, hypothalamus, vagal lobe, brainstem, or retina. However, *noc-ab* transcripts are four times more abundant in the retina than in the telencephalon, diencephalon, hypothalamus, cerebellum, and vagal lobe; meanwhile, the brainstem and optic tectum showed the lowest abundance (Figure 13D). The expression of *noc-bb* is quite similar within the brain, with the highest abundance of transcripts in the diencephalon.

The relative abundance of the three *noc* paralogues in each of the tissues is represented in the Appendix A. The gills, heart, esophagus, intestinal bulb, middle and posterior intestine, spleen, caudal kidney, skin, telencephalon, and hypothalamus showed similar amounts of the three paralogues. However, *noc-bb* is the most abundant paralogue in the head kidney, muscle, and gonads (up to 14 times, Appendix A) and in the optic tectum, vagal lobe, and brainstem (Appendix A). By contrast, *noc-aa* and *noc-ab* transcripts are 2–3 times more abundant than *noc-bb* in the anterior intestine, while *noc-ab* is 2–3 times more highly expressed than *noc-aa* or *noc-bb* in the hepatopancreas and the retina.

The relative abundance of *noc-a* and *noc-b* in the peripheral tissues, brain, and retina of zebrafish is summarized in Figure 14. The *noc-a* transcripts are highly expressed in the liver and pancreas (25–50-fold) compared to the brain and most of peripheral tissues, except for muscle (4-fold) (Figure 14A). The abundance of *noc-b* transcripts is more homogenous among tissues, with the highest amounts in the liver, gonads, muscle, and retina (Figure 14B); intermediate amounts in the pancreas, adipose tissue, and gonads; and significantly lower amount in the rest of the tissues. Regarding the relative expression of *noc-a* vs. *noc-b* in each of the tissues, similar amounts of both paralogues were found in the head kidney, gills, esophagus, posterior intestine, and the spleen (Appendix A). However, the intestinal bulb, anterior intestine, adipose tissue, caudal kidney, gonad, skin, muscle, brain, and retina were enriched in *noc-b* expression, while the *noc-a* was the most expressed paralogue in the liver and pancreas (Appendix A).

## 3. Discussion

This work presents the first complete description of the splicing variants of the different *nocturnin* paralogues in cyprinid fish. To date, there are only two published studies in fish about this enzyme from partial gene sequences. In zebrafish, two isoforms of *noc* (*noc-a* and *noc-b*) in the retina have been characterized [5], establishing the basis for the nomenclature of this gene in the rest of the teleosts. In goldfish, *noc* expression has been related to feeding and fasting in the hypothalamus, hepatopancreas, and intestinal bulb [6].

Our present in silico analysis of *noc* genes from a primitive Osteichthyes, the spotted gar (*Lepisosteus oculatus*), indicates the possibility of an alternative splicing mechanism with three variants (Figure 1: I, II, III), which leads to a high complexity of gene expression, with functional specialization of each of the isoforms. If this hypothetic specialization confers biological advantages, it should be present in other vertebrate groups. Currently, two variants (I and II) can be identified in many genomes of Chondrichthyes, Osteichthyes, and Tetrapoda. However, we can only localize the three variants I, II and III in a few basal Osteichthyes, as coelacanth (*Latimeria chalumnae:* XP_005996687.1, XP_005996691.1, XP_005996689.1) and American paddlefish (*Polyodon spathula:* XP_041133185.1, XP_041133209.1, XP_041133201.1) in addition to spotted gar. Intriguingly, human nocturnin is an exception, with only a splice variant I.

It has been demonstrated that a teleost common ancestor underwent a full genome duplication (3R) 350 million years ago [24,28], generating the duplication of many genes, including *noc* with two paralogues: *noc-a* and *noc-b*. Due to this origin, both *noc-a* and *noc-b* are always found in separate chromosomes. After the additional genomic duplication of the Cyprininae subfamily (4Rc), estimated around 10–15 million years ago [26,29], four *nocturnin* genes were expected in separate chromosomes. We examined genomes from several allotetraploid species of *Sinocyclocheilus*, *Carassius*, and *Cyprinus carpio*, and we found two orthologous genes for zebrafish *noc-a* and two for *noc-b*. Thus, it is important to create a consistent nomenclature for these four paralogues in the Cyprininae subfamily.

The present results report the complete coding sequences (cds) of three *nocturnin* paralogues, allowing us to differentiate between *noc-aa* and *noc-ab* in goldfish. In other cyprinines, there are also two orthologous of the zebrafish *noc-b*, but only the *noc-bb* paralogue has been identified in goldfish by RT-PCR. After the duplication process, several mechanisms may preserve both copies, such as subfunctionalization and neofunctionalization. However, when a duplicated gene acquires a very specialized function, the most likely scenario is that one of the copies would be lost [28]. This is the case for *noc-ba* in the genus *Carassius*, where before total deletion, this gene accumulates deleterious mutations that hinder its transcription and subsequent translation into protein. Searching the complete genome of *Carassius auratus* we have found a sequence with high homology with the *noc-ba* of other cyprinines. However, the sequence shows several deletions in the coding regions (exons 2 and 3) and some insertions (exon 3). Therefore, we suggest that *noc-ba* could be a pseudogene that has accumulated deleterious mutations that changed the reading frame, and subsequently stopped transcription into functional mRNA. In addition, the genome of Prussian carp (*C. gibelio*) shows a highly mutated *noc-ba* (Figure 3), indicating that the process of pseudogenization started in the ancestor of both species.

In addition, to obtain the cds of the three isoforms of nocturnin, in this work, we aim to confirm the expression of the possible splicing variants of each paralogue, following the model of spotted gar *noc*. The common teleost ancestor, prior to 3R duplication, generated three possible transcripts for the first exon by alternative splicing, i.e., variants I, II and III. Some of these splicing variants have been conserved in the teleost evolution, and others have been lost. Our results, based on mRNA sequencing, demonstrate the splicing variant I and II for the two *noc-a* isoforms, (Figure 4 and Figure 5), while II and III variants have been maintained for the *noc-bb* form (Figure 6). Nevertheless, failure to detect the variant III for *noc-aa* and *noc-ab* and the variant I for *noc-bb* may be due to a possible low expression in the sequenced tissue (hepatopancreas). To exclude this possibility, searching in goldfish transcriptomic SRA libraries obtained from several tissues (Appendix A), we found that the most abundant is the variant I for *noc-aa* and *noc-ab*; meanwhile, the forms II and III are a thousand times less abundant. The main variant for *noc-bb* is II, with the form III being minimal. Oddly, we found a very low, but significant, presence of the variant II of pseudogene *noc-ba* in SRA libraries.

Each splice variant needs a different codon ATG as translation initiation sites, but none of them have a canonical Kozak consensus sequence (5′-gccRccATGG-3′, see Figure 4, Figure 5 and Figure 6) This event has been described for mammalian NOC, and several isoforms could be generated from the same mRNA by leaky translation initiation [30].

Comparing our sequences (comet goldfish variety) with those of the *Carassius auratus* (strain Wakin) genome in the NCBI database, we found some differences, mainly SNPs, which cause a few changes in protein sequences, as is expected when comparing sequences from different individuals of the same species. The NOC-AB proteins are identical, while the NOC-BB differs in one amino acid. By contrast, the sequence of NOC-AA seems to be more variable, as we found five different amino acids scattered in the protein sequence, and the specimen we used for sequencing was heterozygotic for *noc-aa*, with two alleles differing in four amino acids (Figure 4).

We note that *noc-aa* from the NCBI genome showed an insertion of two extra Ser at the beginning of variant II, with respect to our sequence WNX29031. This location is a hypervariable region for cyprinine NOC-AA and NOC-AB, with three to seven serial Ser repeats (Appendix A) with an unknown functional meaning. This corresponds to variable tandem repeats of codons AGC or AGT. Insertion or deletion (indels) of repeat triplets is a highly probable event in some genes [31,32].

The well-preserved presence of multiple splicing forms for fish nocturnins makes us look for the possible functional role of each variant. We searched for specific sequence motifs at the N-terminal region. In variant I of mammalian *noc*, there is a putative mitochondrial-targeting signal (MTS) [30,33], which directs the nascent NOC chain into its final mitochondrial location. Once there, NOC suffers an intramitochondrial MTS elimination to release the core catalytic domain of enzyme [34]. The analysis carried out in this report allowed us to confirm that all variant I of cyprinids and most fish contain a real MTS of 76–90 amino acids (Appendix A), with an initial amphipathic α-helix with net positive charge, followed by a disorganized region, and ending with an α-helix containing the appropriate cleavage sites for mitochondrial peptidases.

The variant II is the shortest NOC sequence, and it has a highly conserved N-terminal motif (MGXXXS) in fish. This is the consensus sequence for the N-terminal glycine myristoylation. The co-translational mechanism involves the cleavage of N-terminal methionine from newly formed peptide by methionine aminopeptidases, and the addition of myristoyl group by cytoplasmic N-myristoyltransferases to now free glycine residue. This modification allows interaction with cell membranes and directs the final location of variant II. In mammalian cells, this cytoplasmic form of NOC is enriched on the outer surface of endoplasmic reticulum membranes [30].

The sequence of variant II is well conserved in most vertebrates, but we found a hypervariable region after an initial MG motif in cyprinid *noc-aa* and *noc-ab* (Appendix A). Due to the proximity to myristoylation site, the number of serial serines in this hypervariable region may affect the myristoylation efficiency. Considering scores from Myristoylator tool, we can infer that the optimal serine number is 4 to 5, with scores higher than 0.97. Lower or higher Ser repeats significantly reduced the score. The sequence from goldfish genome (strain Wakin) shows a huge reduction in score (six serines: score −0.3432) with respect to our sequence of *noc-aa II* from the comet variety (WNX29031, four serines: score 0.9867). Then, in addition to the differences among species, there are also differences between specimens with functional consequences.

Variant III has a very short coding exon 1c of only 5–7 amino acids, which makes it difficult to detect by automated annotation of genomes, and only some fish species have it well annotated. Initially, we discard the variant III for *noc-a* paralogues in Cyprinidae, but a variant III of *noc-ab* is annotated in the Prussian carp genome (accession no. XP_052429539.1). A detailed search in the intronic sequences upstream from exon 2 allows us to detect this short exon, located at 2000–3000 bp upstream from exon 2 in all cyprinine species. The presence of this short coding exon overrides the N-myristoylation signal in variant III and creates an N-terminal motif M(D/E) detected by ELM software (Appendix A). This is part of the DEG Nend UBRbox 2 motif, related to its turnover rate by targeting proteins for ubiquitin-dependent proteasomal degradation. The methionine aminopeptidases cleave methionine and expose the negatively charged amino acid (Glu or Asp) to arginylation by arginyl transferases. Then, the positively charged N-Arg peptide is rapidly ubiquitinated and degraded [35].

This splicing variant III may assume a central role in the circadian function of nocturnins, as a putative clock-controlled gene [8]. To accomplish this role, it is required that NOC is a rapidly inducible, short-lived protein. In fact, classical clock-controlled genes, such as arylalkyl N-acetyltransferase, which is the rate-limiting enzyme of the daily rhythm of melatonin, have high turnover rates [36]. The first published NOC sequence (accession no. AAB39495), which was proposed as a clock-controlled gene in *Xenopus laevis* retina [1], is difficult to classify as homologous to variant I, II or III. It has an exon 1 of 21 amino acids with no MTS motifs, but it has a DEG Nend UBRbox 2 with an N-terminal (MD-) motif (Appendix A). Thus, it may be the functional equivalent of a variant III in fish, justifying the fast and huge circadian rhythm of its expression in the frog retina, which shows an increase in mRNA from minimum to maximum value in only 4 hours [1], and a drop in protein from maximum to minimum in 4 hours [12].

For the core catalytic domain, most of it located in exon 3 and present in all splice variants, we verified the conservation of residues involved in Mg^2+^-dependent phosphatase activity for goldfish, zebrafish, human, and frog NOC (Figure 7). Taking human NOC coordinates as reference, we found the conservation of the residues involved in the two Mg^2+^ bindings, for site 1 (Asn^149^, Glu^195^) and site 2 (Asp^324^, Asn^326^, Asp^377^) [27]. Two lysines (Lys^219^, Lys^288^) involved in the binding of a pyrophosphate group from NADP(H) substrate, and two histidines (His^286^ and His^414^ near Mg^2+^ site 2), as catalytic residues, are also conserved [15]. Homologous sites of all these conserved residues are also found in other members of the CCR4 family of the EEP superfamily (Exonuclease, Endonuclease, Phosphatase) [27]. However, we can point to residues only conserved in vertebrate NOC, goldfish included, but not in other CCR4 members, such as Arg^290^ and Arg^367^ (involved in the binding of the adenosine moiety of substrate NADP(H)), which allow the substrate selectivity of NOC [15].

The presence of a leucine-rich repeat motif (LRR) as a component of the multiprotein complex CCR4-NOT, belonging to the same superfamily as NOC [37], has been demonstrated. Its presence has been implicated in mediating protein–protein interactions [38]. In NOC, an LRR motif homologous to CCR4 does not exist [39]. However, we found four leucines in exon 2 with heptad symmetry forming part of a α-helix, characteristic of leucine zipper [40]. The complete conservation is only observed in *Xenopus tropicalis* NOC, although some variations appear in the other analysed species. Nevertheless, despite the variations in this leucine zipper in goldfish NOC sequences, the secondary structure prediction shows an α-helix in that region, indicating that it is possible for NOC to maintain the ability to interact with other proteins [7]. In this sense, mouse NOC binds and promotes the nuclear translocation of PPARγ, but binding is through a nocturnin region far from the leucine zipper [20].

Allopolyploid organisms have an advantage over autopolyploids because chromosomes of patrilineal and matrilineal origin will be maintained as separate and stable evolutionary units (without meiotic recombination between them); therefore, they will maintain fertility despite the initial negative charge of polyploidy. It is necessary to consider that goldfish is an allotetraploid species (4*n* = 100), as are the rest of the cyprinines. The origin of this genomic duplication seems to be a hybridization of two ancestral species from different genera. The matrilineal subgenome (subgenome B) seems to come from a species near *Spinothorax* [29] or *Puntius* [26], while the patrilineal subgenome (subgenome A) comes from a species of tribe Probarbini [26]. Regarding the chronology of these genomic events, and based on the molecular clock, it is assumed that the divergence between the *Danio* genus and the rest of the cyprinines took place 44 million years ago, the divergence between patrilineal and matrilineal ancestors of cyprinines 15 million years ago, and the process of allotetraploidization 13.7 million years ago [29].

Due to the recent origin of the allotetraploid cyprinids, it is likely that the sequences have not accumulated many differences, meaning it is not possible to determine a classification based only on sequence similarity. Therefore, we used two criteria to develop our classification: a phylogenetic tree based on coding sequences, and a synteny conservation analysis. These two criteria have shown a high phylogeny–synteny correlation for all *nocturnin* paralogues in the different species studied.

The chromosome naming of *Cyprinus carpio* and *Carassius gibelio* obeys the allotetraploid origin; they are divided in two sets of 25 chromosomes (A1 to A25, and B1 to B25 following the division of genome in the subgenomes A and B). The *Carassius auratus* genome too, although its chromosome nomenclature is different, has matrilineal chromosomes from 1 to 25, and a patrilineal subgenome from 26 to 50 (e.g., a pair of 4Rc ohnologue chromosomes are 1 and 26).

We expect that the four paralogues, *noc-aa*, *noc-ab*, *noc-ba* and *noc-bb*, will be located on goldfish chromosomes 39, 14, 26 and 1, respectively. However, we identified some deficiencies in the goldfish chromosome numeration: *noc-aa* is on an unplaced segment (NW_020527513, Figure 9) due to the incomplete assemblage of chromosome 39, but with a high homology with the corresponding patrilineal A14 chromosome from *C..carpio* and *C. gibelio*. In addition, following the expected orthology, goldfish *noc-bb* should be in the matrilineal chromosome 1, and *noc-ba* in the patrilineal chromosome 26, but their actual locations are the opposite. This mistake may be due to the wrong naming of chromosomes 1 and 26, or to a less probable ohnologous reciprocal translocation of the *noc*-containing segment.

Many methods have been published to quantify synteny, e.g., [41,42]. These bioinformatics tools are designed to analyze whole genomes in order to obtain large-scale synteny mapping based on small synteny blocks. However, the complexity of these tools is likely to yield widely divergent results, and better and simpler tools [43] are needed to achieve our objectives.

Based on synteny maps, we proposed two new indicators to quantify the degree of synteny conservation of neighboring *noc* genes by comparing two related species. The synteny index (SI) is useful for determining how many gene positions are conserved in a chromosomic fragment; we have found values higher than the threshold level. Moreover, the synteny conservation rate (SCR) reveals a conservation higher than the 50% of goldfish synteny with respect to zebrafish, common carp, and Prussian carp. Values obtained in these indicators match the phylogenetic tree topology (Figure 11C). In addition, these two numerical parameters may be used to confirm the correct assignment of a gene to its orthologous group. When these parameters are calculated in the ohnologous contrast (matrilineal vs. patrilineal chromosomes), the values drop considerably, indicating the absence of a real orthology between these ohnologous chromosomes (Appendix A).

The phylogenetic tree of protein sequences (Figure 12) supports and complements the conclusions of the synteny analysis. The tree in the present report follows the phylogenetic relationships of vertebrates in general, and teleosts in particular. It identifies the position or 3R teleostean genome duplication that generates the NOC-A and NOC-B branches. Furthermore, the 4Rc duplication segregates the cyprinine nocturnins in matrilineal and patrilineal paralogues.

We have included this phylogenetic information in the rationale of nocturnin nomenclature. We add to the zebrafish-based name a letter (a or b), which indicates the inclusion in patrilineal or matrilineal subgenomes, respectively. These criteria may be used for naming the rest of the cyprinine genes, and even extended to other allopolyploid groups. Then, the last letter of all gene names on a chromosome must be a or b, in reference to which subgenome the chromosome belongs. Cyprinidae underwent multiple and independent allopolyploidizations [44], but the same rationale can be extrapolated to other tetraploid groups, just by changing the last letter of gene names for another (different from a or b), indicating different ancestral subgenomes. Thus, the allotetraploid clawed frog *(Xenopus laevis)* separates the genome in S and L subgenomes [45].

Current knowledge regarding the physiological actions of NOC in fish is very limited. The ubiquitous tissue expression of all *noc* paralogues in zebrafish and goldfish, in agreement with previous observations of the expression pattern of *noc-a* and *noc-b* [6], argues in favor of its involvement in many different physiological functions. It is important to consider that primers used for the quantification in the previous study in goldfish [6] were different from the ones used in the present report, and showed the total amount of both *noc-aa* and *noc-ab* summed together. Nevertheless, only one discrepancy is found in the two goldfish studies, which concerns the pituitary. The highest expression of *noc-aa* and *noc-ab* is shown in Figure 13, but a very low expression of total *noc-a* is described in the previous paper [6]. Present results in the brain allow us to conclude that the three paralogues are equally abundant in the forebrain, while the *noc-bb* paralogue is the most expressed in the mid- and hindbrain. This enriched expression of *noc-bb* in the optic tectum, vagal lobe and brainstem could be due to selective induction of this paralogue or repression of the other paralogues. Consistently, *noc-b* expression is three times that of *noc-a* in the brain of zebrafish, reinforcing the relevance of this paralogue. This wide distribution of *noc* in brain areas is also described in mice with different daily profiles of expression [7]. It is plausible to hypothesize a relationship between *noc*, as a clock-controlled gene, and the metabolic requirements of different brain areas [7]. Further studies exploring the rhythmicity of *noc* paralogues in specific brain areas of fish are needed.

All the *noc* paralogues are highly expressed in the retina of both goldfish and zebrafish, but *noc-ab* is more abundant in the subfamily Cyprininae, while *noc-b* is more abundant in Danioninae. This result could indicate an exchange of functional specialization of these paralogues since the divergence of these two cyprinid subfamilies. In zebrafish retina, the presence of *noc-a* transcripts is also detected (being ten times lower than *noc-b* transcripts), which could suggest some possible cooperative roles of all the *noc* for adequate retinal physiology. However, the only previous published study failed to detect the presence of *noc-a* transcripts in zebrafish retina [5], which could be explained by the different sensitivities of the experimental approaches in both studies. The cellular localization of *noc* expression in the retina is also controversial. In the clawed frog, *noc* expression is restricted to the photoreceptor cell layer [1]. Meanwhile, in mice, as in zebrafish, *noc* expression is high in photoreceptors, and lower (but clearly detected) in inner retina [46].

The peripheral distribution of *noc* paralogues is quite similar in zebrafish and goldfish, with some interesting exceptions. The broad expression of all paralogues throughout the entire digestive tract of goldfish and zebrafish supports the involvement of NOCs in several digestive processes in teleosts. Similar expression levels of all paralogues are found in the intestinal bulb of goldfish, and a brief fasting upregulated the expression of *noc-b*, but not *noc-a* paralogues [6]. This result and the highest expression of *noc-b* in the intestinal bulb of zebrafish (present results) could assign specialized functions for NOC-B in this region of digestive tract. The high amounts of *noc* transcripts in the hepatopancreas of goldfish and liver of zebrafish found in the present study agrees with results in mammals [7], and support their possible role in the hepatic lipid metabolism. Without physiological evidence and based only on their relative abundance, a more relevant involvement of NOC-A in teleosts, and particularly the NOC-AB paralogue in Cyprininae, could be suggested.

The adipose tissue shows the highest expression of all paralogues in goldfish but a moderate expression in zebrafish, with a significantly higher abundance of *noc-b* than *noc-a*. This result supports a general role for NOC in adipogenesis in Cyprininae, with NOC-B being the most relevant in Danioninae. A similar interspecific pattern is found in the anterior intestine, with enriched expression of both *noc-a* paralogues in goldfish but only of *noc-b* in zebrafish. A functional divergence of NOC proteins in the regulation of intestinal lipid absorption might be suggested, assumed of NOC-A in Cyprininae and of NOC-B in Danioninae. Finally, all the *noc* paralogues are present in the muscle, gonad, skin, and head kidney in both goldfish and zebrafish, but there is no proposed function for NOC in these tissues.

## 4. Materials and Methods

### 4.1. Animal Maintenance

Goldfish (comet goldfish, *Carassius auratus auratus*) and zebrafish (*Danio rerio*, genotype Tübingen) with a body weight of 15.1 ± 2.5 g and 0.9 ± 0.2 g, respectively, were obtained from a local commercial supplier (Industrias Canarias del Acuario S.A., ICA, Madrid, Spain). Goldfish were kept in 60 L aquaria (*n* = 6–7/aquarium) with filtered fresh water at 22 ± 1 °C, and zebrafish were maintained in 4 L racks (*n* = 5/rack) at 27 ± 1 °C. All fishes were under a 12 h light–12 h darkness cycle (12L:12D photoperiod, lights on at 08:00 a.m., i.e., Zeitgeber Time 0, ZT0) with continuous aeration. A commercial flake diet (1.5% body weight, Sera Pond for goldfish, Tropifish ICA for zebrafish) was offered daily at 10:00 h (Zeitgeber time-2, ZT2) for at least three weeks before the assays. Fish were killed with an overdose of anesthesia (MS-222, Sigma-Aldrich, Madrid, Spain) at ZT4, and immediately sampled. All procedures were approved by the Animal Experimentation Committee of Complutense University and the Community of Madrid (PROEX 170.6/20), and according to the Guidelines of the European Union Council (2010/63/EU) and the Spanish Government (RD53/2013) for the use of animals in scientific research.

### 4.2. mRNA Sequencing of Goldfish Nocturnins

To obtain the full-length sequences, we used one 24 h fasted goldfish sacrificed at ZT4, and the whole hepatopancreas was collected.

Total mRNA was extracted using TRI Reagent (Merck, Madrid, Spain) following the manufacturer’s instructions in mechanically homogenized hepatopancreas samples. Then, mRNA samples were treated with RQ1 RNase-Free DNase (Promega, Madison, WI, USA) according to a 4 μg mRNA/1 μL DNase ratio. Then, an aliquot of 1 μg of total RNA was retrotranscripted into cDNA in a 25 μL reaction volume using random primers (Invitrogen, Waltham, MA, USA), RNase inhibitor (Promega) and SuperScript II Reverse Transcriptase (Invitrogen). The reverse-transcription reaction consisted of 25 °C for 10 min, an extension of 50 min at 42 °C, and a denaturalization step at 70 °C for 15 min. The first strand cDNA fragments obtained were used as a template to amplify goldfish sequences with different primers (Merck), which were specifically designed (Appendix A).

The PCRs were performed in a 50 μL reaction volume containing 0.25 μL of HotStart DNA Polymerase recombinant 100 U, 10 μL PCR Buffer (5×, pH 8.5), 4 μL of MgCl_2_ 25 mM (Promega), 1 μL of dNTP mixture 0.3 mM (Invitrogen), and 1 μL of each forward and reverse primer 10 μM and 2 μL of cDNA. The reaction conditions consisted of an initial denaturation at 94 °C for 3 min, followed by 40 cycles of 94 °C for 45 s, 57 °C or 60 °C for 30 s, and 72 °C for at least a 1 min/1 kb size sequence, with a final extension step at 72 °C for 10 min. For some reactions, touchdown PCRs were performed. PCR products were electrophoresed on a 0.5–2% agarose gel. Single bands for each PCR were purified using a GenElute™ Gel Extraction Kit (Sigma-Aldrich) and sequenced using the Sanger dideoxy method (Secugen, Madrid, Spain).

To obtain the consensus sequences for each paralogue, firstly each amplicon was validated as a *noc* sequence with the BLAST 2.14.0 program (http://blast.ncbi.nlm.nih.gov/ accessed on 1 September 2023). All the *noc* sequences accepted as true were entered into the CAP3 program [47] (http://doua.prabi.fr/software/cap3 accessed on 1 September 2023), and the resulting assembled sequence was accepted as consensus when a minimum of five independent sequenced amplicons were obtained for each bp position. Following this procedure, we detected two allelic variants in *noc-aa*.

To obtain the noc full-length sequences, and resolve the two allelic sequences for noc-aa, we used the 4-TOPO cloning kit for sequencing (K4575-02, Invitrogen). The full sequences of *noc* paralogues’ splicing variants were amplified using specific primers for each. After performing the PCR, amplicons were purified. To insert the *noc* sequences into the plasmids, 4 μL of amplified cDNA was incubated at 4 °C for at least 12 h with 1 μL of TOPO vector and 1 μL of salt solution from the cloning kit. Then, 5 μL of plasmid cDNA was added into a tube of chemically competent *E. coli* of the cloning kit and incubated for 30 min at 4 °C, followed by a thermal shock of 45 s at 42 °C to facilitate the bacteria transformation. After that, 800 μL of SOC medium was pipetted into the E. coli tubes and incubated at 37 °C and 600 rpm for 1 h. Finally, different volumes of the primary culture were inoculated in Petri dishes with LB-agar, X-Gal and IPTG (Merck) for at least 24 h. Afterwards, transformed colonies were selected to perform a single subculture in LB medium at 37 °C. Then, a new PCR using M13 primers (Merck) was performed to amplify the complete inserted sequences of plasmids, and their sizes were checked via electrophoresis in 1% agarose gel. Then, 24 h later, the plasmid cDNA was isolated from the subculture tubes (following the manufacturer’s instructions) and sequenced using T3 and T7 primers (Secugen, Santa Clara, CA, USA). A minimum of 20 positive colonies were sequenced for each gene to validate the sequence obtained from cloning with the sequence obtained from PCR-CAP3 consensus.

### 4.3. Blast Analysis of mRNA SRA Libraries

To find the expression of different splicing variants of *noc* in the mRNA Sequence Read Archive (SRA) libraries, we selected SRA projects of *Carassius auratus* obtained from multiple organs, tissues, and developmental stages, available from NCBI databases. We designed a query sequence for each noc variant, consisting of 35 bp upstream and 35 bp downstream from the splicing point between exon 1 and exon 2. For finding matches, we used the Blastn 2.14.0 tool (http://blast.ncbi.nlm.nih.gov/ accessed on 1 September 2023). For positive hit criteria, we accepted a minimum of query coverage of 66% and identity percentage of 96%. With these criteria, false positives are not counted, but SNPs (corresponding to allelic variants) are included.

### 4.4. In Silico Structural Analysis and Phylogenetic Tree Construction

An in silico structural analysis of the *noc* sequences obtained was carried out using online bioinformatics tools. The predictions of molecular weight and other basic properties of the protein were carried out using the ProtParam tool from Expasy (http://web.expasy.org/protparam/ accessed on 1 September 2023). The secondary structure prediction was performed using JPred4 (www.compbio.dundee.ac.uk/jpred/index_up.html accessed on 1 September 2023) [48]. JPred is a web server that uses the PSI-BLAST algorithm to generate secondary structures from the protein sequence. The tertiary structure was made using SWISS-MODEL (https://swissmodel.expasy.org/ accessed on 1 September 2023) [49]. For searching MTS motifs, we used the MitoFates (http://mitf.cbrc.jp/MitoFates/cgi-bin/top.cgi accessed on 1 September 2023) [50], and TPpred 3.0 (https://tppred3.biocomp.unibo.it accessed on 1 October 2023) [51] tools. N-myristoylation probability is calculated with Myristoylator (https://web.expasy.org/myristoylator/ accessed on 1 September 2023) [52], as assuming a high confidence in myristoylation when the score was higher than 0.85. Other molecular motifs were analyzed with the Eukaryotic Linear Motif (ELM) resource (http://elm.eu.org/ accessed on 1 September 2023) [53]. Protein construction from WGS sequences was performed using the Genewise tool from EMBL-EBI (www.ebi.ac.uk/Tools/psa/genewise/ accessed on 1 September 2023) [54].

A phylogenetic analysis was performed by aligning the goldfish NOC amino acid sequences with those of other vertebrates retrieved from GenBank using the Clustal-X2.1 tool (www.clustal.org accessed on 1 September 2023) [55]. The evolutionary model and the phylogenetic tree were constructed with the help of MEGA 11 software [56]. After testing several phylogenetic methods, we opted for maximum likelihood (ML) with 1000 replicates for the bootstrap test. The representation was achieved using NjPlot 2.3 software (http://pbil.univ-lyon1.fr/software/njplot.html accessed on 1 September 2023) [57], considering *Homo sapiens*, *Xenopus tropicalis* and *Latimeria chalumnae* as outgroup species.

### 4.5. Synteny Analysis 

The synteny analysis of *noc* was accomplished on the basis of the most recent assemblies of genomes published for the studied species (Appendix A). Two different websites (www.ncbi.nlm.nih.gov; www.ensembl.org accessed on 1 September 2023) published automatic (computer-based) annotation of predicted genes. These annotations have important differences, as they use different software resources, even though they are based on the same assembly project.

The graphic representation of synteny analysis was drawn using the automatically generated schemes from Genomicus 110.01 (www.genomicus.bio.ens.psl.eu/genomicus accessed on 1 September 2023) [58] as a draft copy, which uses genomic information from the Ensembl database. Furthermore, we manually curated the graphics by checking gene per gene in all the species considered, comparing the Genomicus synteny maps with the NCBI sequences, to confirm the loci analyzed and their conservation.

The nomenclature of the predicted genes was also reviewed, and the most suitable name was verified using the BLAST tool by comparing their sequences with the GenBank sequences that showed higher homology. Additionally, the ZFIN database (www.zfin.org accessed on 1 September 2023) [59] was also used for the gene nomenclature of *Danio rerio*.

For the final graphic representation, the scheme of the analyzed chromosomes was arranged in parallel, and each gene is represented by a pentagon whose sharp vertex indicates the sense of transcription on the chromosome (right: the coding strand is the direct one; left: the coding strand is the reverse complement). Orthologous genes are represented in the same color and are joined by straight lines to facilitate their location. In the center of the maps Is the *Danio rerio* chromosome (with a white background), which has been used as a reference, and over that, there are the matrilineal chromosomes (with a pink background) in the following order: *Sinocyclocheilus grahami*, *Cyprinus carpio*, *Carassius gibelio*, and *Carassius auratus*. Towards the bottom, the patrilineal chromosomes (with a cyan background) are arranged in the corresponding symmetrical order: *Sinocyclocheilus grahami*, *Cyprinus carpio*, *Carassius gibelio*, and *Carassius auratus*.

To carry out a quantitative synteny analysis and the assessment of the synteny conservation for any gene cluster, two new parameters were determined: the synteny index (SI) and the synteny conservation rate (SCR) (Appendix B).

### 4.6. Tissue Expression of noc mRNA Using RT-qPCR

Samples of the telencephalon, diencephalon, optic tectum, hypothalamus, pituitary, cerebellum, vagal lobe, brainstem, retina, head kidney, gills, heart, esophagus, intestinal bulb, anterior intestine, middle intestine, posterior intestine, hepatopancreas, spleen, kidney, gonads, adipose tissue, skin, and muscle were collected from 6 goldfish at ZT4.

To analyze the mRNA distribution of *noc* in zebrafish, brain, retina, head kidney, gills, heart, esophagus, intestinal bulb, anterior intestine, posterior intestine, liver, pancreas, spleen, kidney, gonads, adipose tissue, skin, and muscle were collected from 5 fish at ZT4.

Total mRNA extraction and reverse-transcription of 2 μg of RNA were performed as described above. Then, real-time quantitative PCR (RT-qPCR) was performed using iTaq Universal SYBR Green Supermix (Bio-Rad, Hercules, CA, USA). The specific primer sequences used for the target genes (Appendix A) were designed to amplify *noc-aa, noc-ab* and *noc-bb* specifically. The primers for goldfish were hand-made from full cds sequences obtained in the present work. Primers for the reference genes of goldfish (*β-actin* and *ef-1α*) [60] and for zebrafish nocturnins, and the reference gene (*β-actin*) were previously described [5,61].

Genes were amplified using 1 μL of cDNA and 0.5 μL of each forward and reverse primer 10 μM in a final volume of 10 μL. Each PCR run included a standard curve of cDNA and a negative control (water). The qPCR cycling conditions consisted of a ramp of 95 °C for 30 s and 40 cycles of a two-step amplification program (95 °C for 5 s and 60 °C for 30 s), except in the case of goldfish *noc-bb*, wherein an additional step of 58°C for 30 s was added to the amplification program. The melting curve was systematically monitored (with a temperature gradient of 0.5 °C/5 s from 65 to 95 °C) at the end of each run to confirm the specificity of the amplification reaction. The 2^−ΔΔCt^ method was used to determine the relative mRNA expression [62], considering a the relative value of “1” for the brain in zebrafish and the hepatopancreas and diencephalon in the peripheral and neural regions of goldfish, respectively.

### 4.7. Statistics

Statistic analyses were carried out using SigmaPlot 12.0 (Systat Software Inc., San José, CA, USA). Data were checked for normality (Shapiro–Wilk test) and homoscedasticity (Levene’s test). Statistical differences in goldfish *noc-aa, noc-ab*, and *noc-ba*, or zebrafish *noc-a* and *noc-b* expression among different tissues were analyzed using a one-way ANOVA, followed by the post hoc Student–Newman–Keuls (SNK) multiple comparisons test. Significant differences were considered at *p* < 0.05.

## 5. Conclusions

This work presents the first full characterization of molecular structure of the different *nocturnin* paralogues in fish. The only human *noc* gene generates two (mitochondrial and cytoplasmic) isoforms of protein NOC through alternative translation initiation sites of a single mRNA, while in fish, the molecular diversity of NOC is achieved by whole genome duplication and by alternative splicing.

The *noc* gene of basal Osteichthyes has three splicing variants I, II and III, while in many genomes of Chondrichthyes, Osteichthyes, and Tetrapoda, only two variants (I and II) can be identified. From the four expected paralogues in goldfish, by direct mRNA sequencing, we obtained two isoforms for the *noc-a* subgroup (*noc-aa* and *noc-ab*) with two splice variants (I and II) each, and only one for the *noc-b* subgroup (*noc-bb*) with two transcripts (II and III). In genus *Carassius*, the other expected paralogue *noc-ba* is a pseudogene.

The splicing variant I of cyprinids and most fish contains an MTS motif that predestines this NOC variant to be a mitochondrial enzyme. Variant II has a conserved N-terminal glycine myristoylation motif followed by a hypervariable region with several serial serines that affect its myristoylation efficiency; it is species- and specimen-specific. Variant III has a very short coding exon 1c containing a DEG Nend UBRbox 2 motif, which means this variant is targeted for rapid ubiquitin-dependent proteasomal degradation. The core of the catalytic domain conserves the residues involved in Mg^2+^-dependent phosphatase activity in all splicing variants.

From the synteny analysis and the phylogenetic tree of protein sequences, we proposed a classification and a long-lasting nomenclature of *noc* in goldfish extrapolated to allotetraploid Cyprininae, based in the localization of *noc* genes in the patrilineal or matrilineal subgenomes. We add to the zebrafish-based name a letter (a or b), indicating their inclusion in patrilineal or matrilineal subgenomes, respectively.

All the *noc* paralogues are ubiquitously expressed in goldfish and zebrafish, which support their possible involvement in several physiological functions. However, there are also interspecies differences regarding the relative abundance of the *noc* paralogues in several tissues.

## Figures and Tables

**Figure 1 ijms-25-00054-f001:**
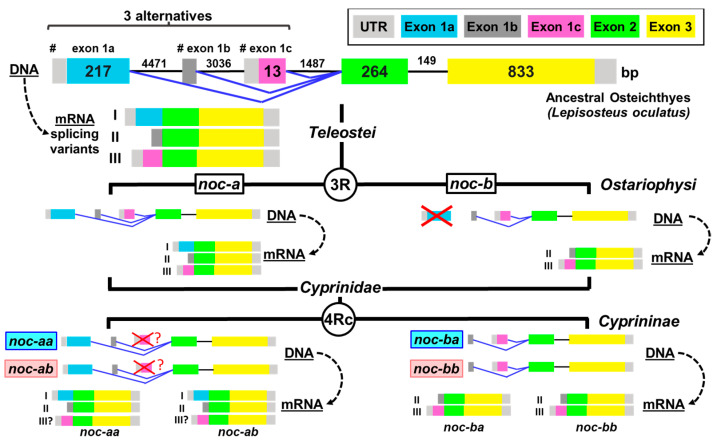
*Nocturnin* gene structure in Osteichthyes. Full-genome duplications are indicated as 3R (Teleostei specific) and 4Rc (Cyprininae specific). Proposed model for the exon–intron structure of the gene encoding nocturnin in different Osteichthyes clades, and its transcription into mRNA (dashed lines). The I, II and III indicate the alternative splicing variants. # indicates the alternative transcription start sites. Exons are represented by boxes and introns by lines. Angled blue lines indicate alternative splicing of the first exon (exon 1a, exon 1b or exon 1c) in mature mRNA. The length (bases pair, bp) of exons (inside the boxes) and introns (above lines) is indicated for the *Lepisosteus oculatus* gene. Lost exons during evolution of Ostariophysi and Cyprininae are represented by red crosses. Question marks indicate unconfirmed loss of variant III in the *noc-a* paralogues of cyprinines. Light pink and cyan backgrounds on Cyprininae nocturnin names indicate matrilineal and patrilineal origin, respectively.

**Figure 2 ijms-25-00054-f002:**
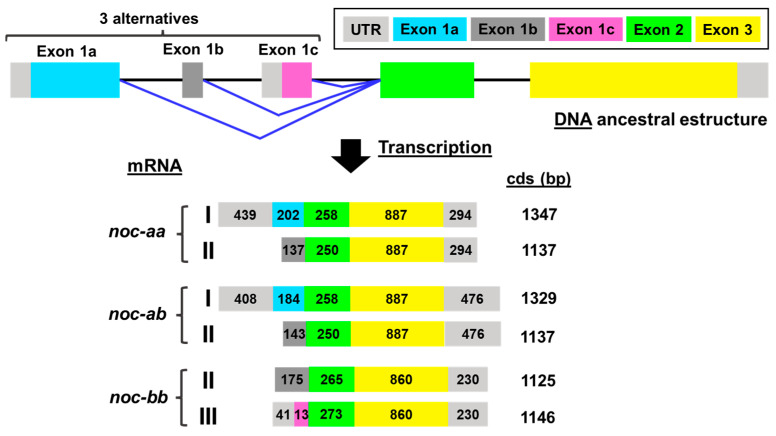
Exon-intron structure and transcription pattern of *nocturnin* gene paralogues in goldfish obtained by mRNA sequencing. I, II, and III indicate the alternative splicing variants. Exon 1a, exon 1b, and exon 1c are the alternatives of the first exon in mature mRNA. Angled blue lines indicate an alternative splicing mechanism. Coding exons are indicated by colored boxes and introns by lines. Grey boxes indicate untranslated regions (UTR and exon 1b). The length (bases pair, bp) of exons is indicated inside the boxes. The length of coding sequences (cds) is also indicated.

**Figure 3 ijms-25-00054-f003:**
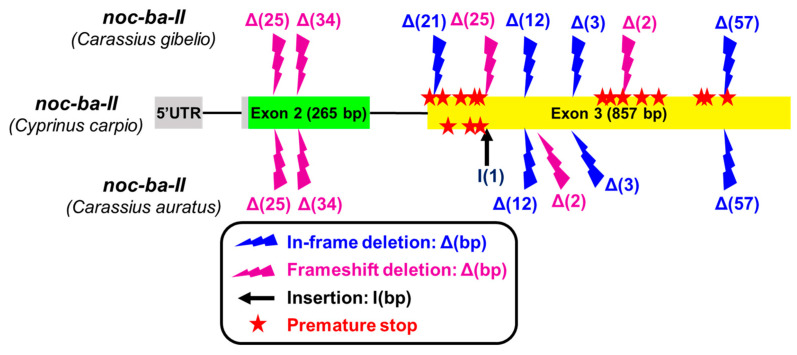
Mutations accumulated in the *noc-ba* paralogue of *Carassius gibelio* (Gene ID: 127988217) and *Carassius auratus* (Gene ID: 113105808) compared to *Cyprinus carpio* (accession no. XM_019094354.2). Illustration represents the exon–intron structure of the splicing variant II of *noc-ba*. Exons are indicated by boxes and introns by lines. The symbols above (*C. gibelio*) and below (*C. auratus*) indicate the mutations compared to *C. carpio* sequence. The blue and purple rays represent in-frame and frameshift deletions, respectively, and the base pairs suppressed, Δ(bp). The black arrow I(1) indicates an insertion of 1 bp. The red stars represent premature stop codons in transcribed sequence.

**Figure 4 ijms-25-00054-f004:**
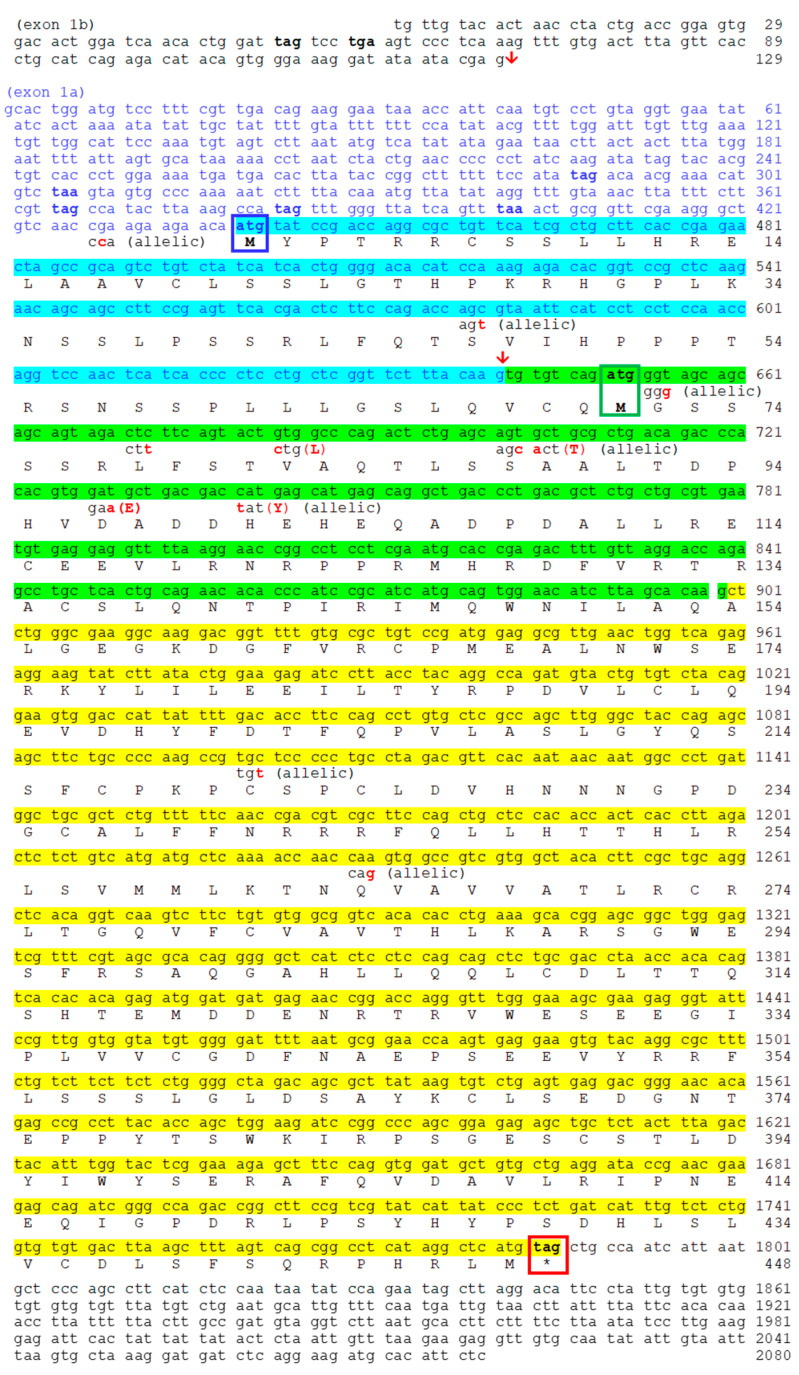
Nucleotide and deduced amino acid sequences of goldfish *noc-aa*. Sequences are accessible through GenBank. Splice Variant I is composed of exon 1a in blue (coding sequence with cyan background), exon 2 (green background) and exon 3 (yellow background) (allele 1 accession no. OR651354). Splice Variant II is composed of non-coding exon 1b, exon 2, and exon 3 (accession no. OR651356). Red arrow indicates the insertion point of exon 1a or exon 1b on exon 2. Bold font indicates stop codons upstream of the initial methionine codon. Coding region extends from the first methionine residue (blue box for variant I, green box for variant II) to the stop codon (* and red box). Backgrounds colored cyan, green, and yellow indicate the coding part of exon 1a, 2, and 3, respectively. Red letters indicate differences in nucleotides and amino acids between the two allelic sequences (allele 2, accession no. OR651355 and OR651357).

**Figure 5 ijms-25-00054-f005:**
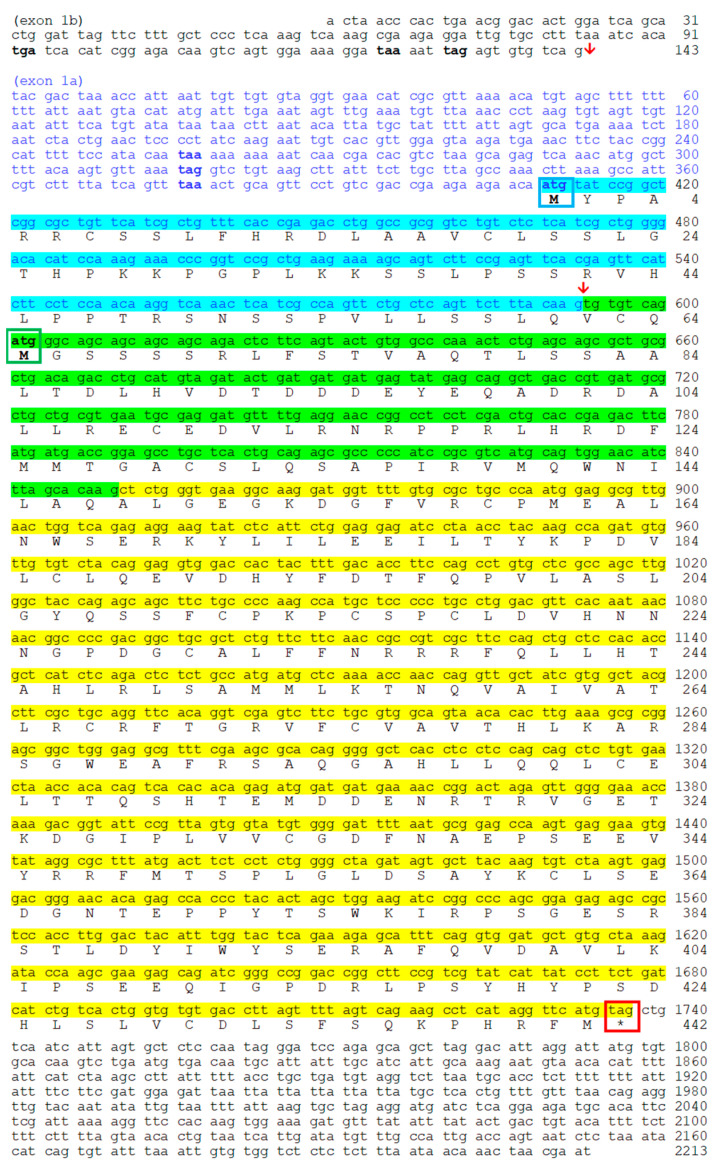
Nucleotide and deduced amino acid sequences of goldfish *noc-ab*. Both sequences are accessible through GenBank. Splice Variant I is composed of exon 1a in blue (coding sequence with cyan background), exon 2 (green background), and exon 3 (yellow background) (accession no. OR651297). Splice Variant II is composed of non-coding exon 1b, exon 2, and exon 3 (accession no. OR651298). Red arrow indicates the insertion point of exon 1a or exon 1b on exon 2. Bold font indicates stop codons upstream of initial methionine codon. Coding region extends from the first methionine residue (blue box for variant I, green box for variant II) to the stop codon (* and red box). Backgrounds cyan, green and yellow indicate the coding part of exons 1a, 2, and 3, respectively.

**Figure 6 ijms-25-00054-f006:**
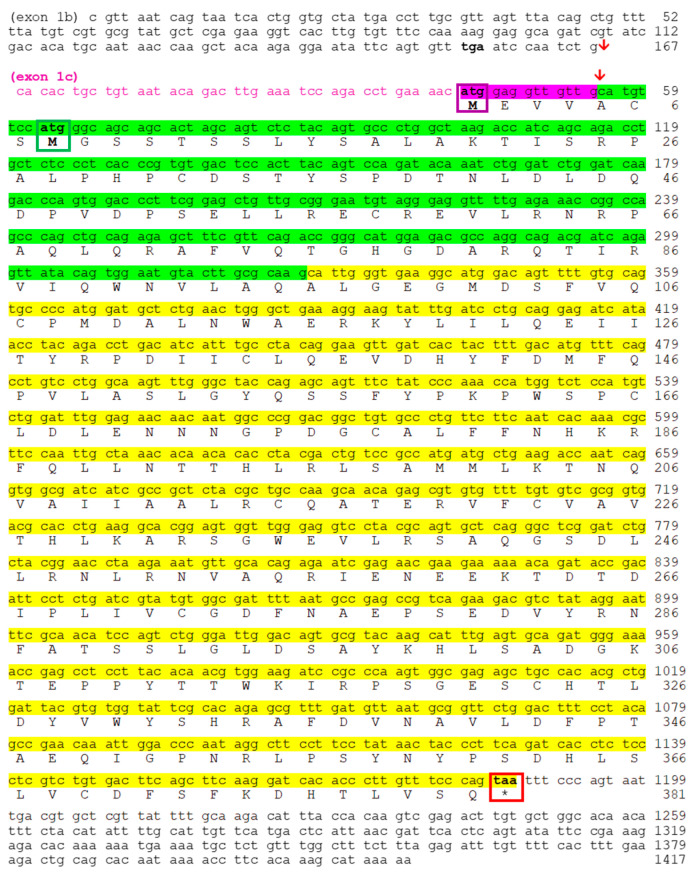
Nucleotide and deduced amino acid sequences of goldfish *noc-bb*. Both sequences are accessible through GenBank. Splice Variant III is composed of exon 1c in purple (coding sequence with purple background), exon 2 (green background), and exon 3 (yellow background) (accession no. OR651299). Splice Variant II is composed of non-coding exon 1b, exon 2, and exon 3 (accession no. OR651300). Red arrow indicates the insertion point of exon 1b or exon 1c on exon 2. Bold font indicates stop codons upstream of initial methionine codon. Coding region extends from the first methionine residue (purple box for variant III, green box for variant II) to the stop codon (* and red box). Backgrounds colored purple, green and yellow indicate exons 1c, 2, and 3, respectively.

**Figure 7 ijms-25-00054-f007:**
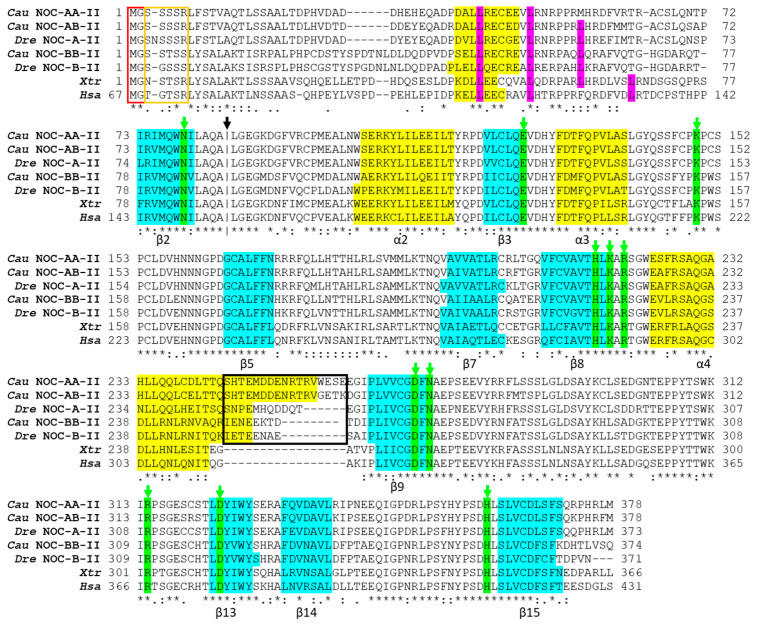
Alignment of the deduced amino acid sequences of splicing variant II of goldfish NOC-AA (Cau NOC-AA-II: WNX29031), NOC-AB (Cau NOC-AB-II: WNX29026), and NOC-BB (Cau NOC-BB-II: WNX29028) with NOC sequences from *Danio rerio* (Dre NOC-A-II: XP_005169676.1) and (Dre NOC-B-II: XP_021331689.1), *Xenopus tropicalis* (Xtr: KAE8630336.1), and *Homo sapiens* (Hsa: NP_036250.2). Multiple sequence alignment was conducted using Clustal v.X2. Bottom symbols (. : *****), indicating a significant conservation from minor to major. Horizontal hyphens indicate gaps introduced to optimize the alignment. The vertical hyphen line (black arrow) marks the transition between exon 2 and 3. The secondary structure was predicted using JPred4. Structural elements are indicated as follows: yellow boxes (α-helix) and blue boxes (β-sheet). The names of α and β follow the nomenclature of Abshire and coworkers [27]. Green boxes pointed to by a green arrow indicate the amino acids conserved in the active site. Pink boxes mark the position of the heptad leucine repeat. Red rectangles indicate putative myristoylation sites, and the orange rectangle indicates a Ser/Thr-rich region. Black rectangles indicate the non-conserved domain of goldfish NOC paralogues compared to human and frog NOC.

**Figure 8 ijms-25-00054-f008:**
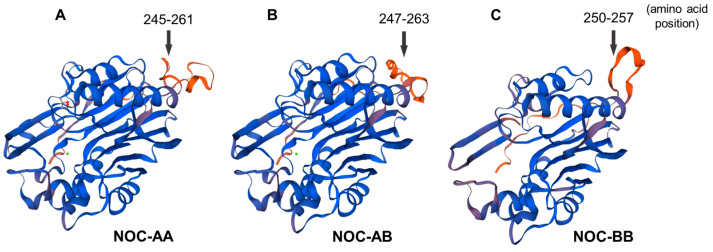
Tertiary predicted 3D structures of splicing variant II of goldfish; nocturnin: NOC-AA (**A**), NOC-AB (**B**), and NOC-BB (**C**). In blue are the conserved segments. The non-conserved domains in human and goldfish are represented in orange (the arrows indicate the amino acids’ position in Figure 7).

**Figure 9 ijms-25-00054-f009:**
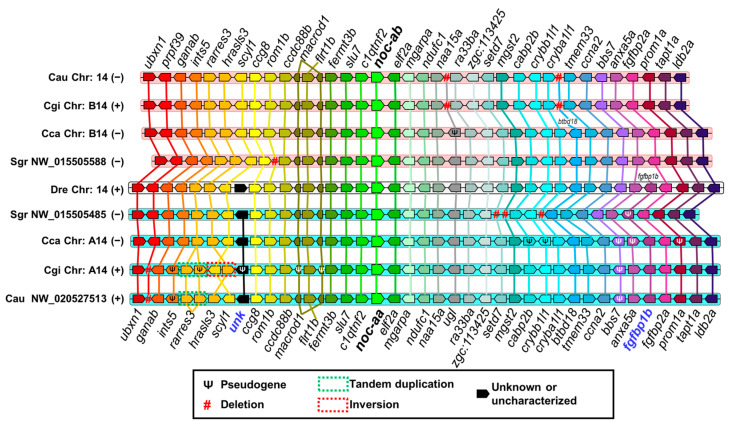
Synteny analysis of *noc-a* paralogues in *Danio rerio* (Dre), *Sinocyclocheilus graham* (Sgr), *Cyprinus carpio* (Cca), *Carassius gibelio* (Cgi), and *Carassius auratus* (Cau). Black bolded names indicate nocturnin paralogues in the center of synteny. Chromosomes are represented with a white background in *Danio rerio* (as the reference species) and a cyan or pink background in the other species, depending on the patrilineal or matrilineal subgenome, respectively, of the ancestral duplication of Cyprininae (4Rc). Chromosome coding corresponds to that of the genome project (Appendix A), and the (+) or (−) symbols represent the sense or antisense orientation of the chromosome, respectively, in the projects analyzed. Genes are represented with pentagons whose sharp vertex indicates the transcription sense. Same colors indicate orthologous genes. Black pentagons represent unknown genes or genes without orthologous in any chromosome analyzed. Mutations are inferred comparing the orthologous chromosomes: pseudogene (Ψ), deletion (#), tandem duplication (green dashed boxes), and inversion (red dashed boxes). Gene abbreviations are indicated in the Appendix A. Gene names in bold blue indicate diagnostic markers of patrilineal chromosomes.

**Figure 10 ijms-25-00054-f010:**
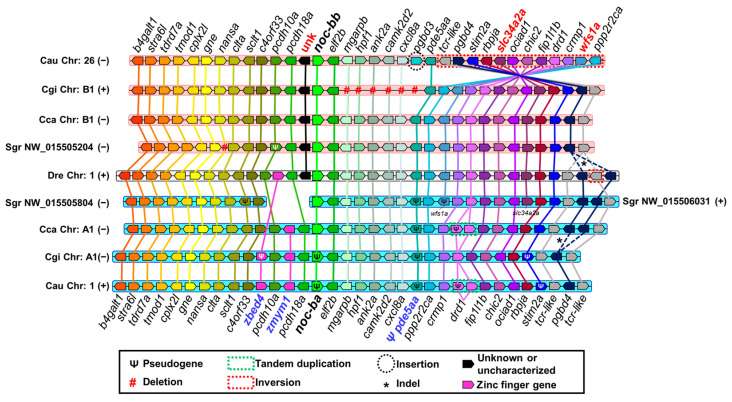
Synteny analysis of *noc-b* paralogues in *Danio rerio* (Dre), *Sinocyclocheilus graham* (Sgr), *Cyprinus carpio* (Cca), *Carassius gibelio* (Cgi), and *Carassius auratus* (Cau). Black bolded names indicate nocturnin paralogues in the center of synteny. Chromosomes are represented with a white background in *Danio rerio* (as reference species) and with a cyan or pink background in the other species depending on the patrilineal or matrilineal subgenome, respectively, of the ancestral duplication of Cyprininae (4Rc). Chromosome coding corresponds to that of the genome project (supplementary Appendix A), and the (+) or (−) symbols represent the sense or antisense orientation of the chromosome, respectively, in the projects analyzed. Genes are represented with pentagons whose sharp vertex indicates the transcription sense. Same colors indicate orthologous genes. Black pentagons represent unknown genes. Pink pentagons indicate zinc finger genes. Mutations are inferred comparing the orthologous chromosomes: pseudogene (Ψ), deletion (#), tandem duplication (green boxes), inversion (red boxes), insertion (black circle), and provisional indel (*). Gene abbreviations are indicated in the Appendix A. Gene names in bold blue (below) or bold brown (above) indicate diagnostic markers of patrilineal or matrilineal chromosomes, respectively.

**Figure 11 ijms-25-00054-f011:**
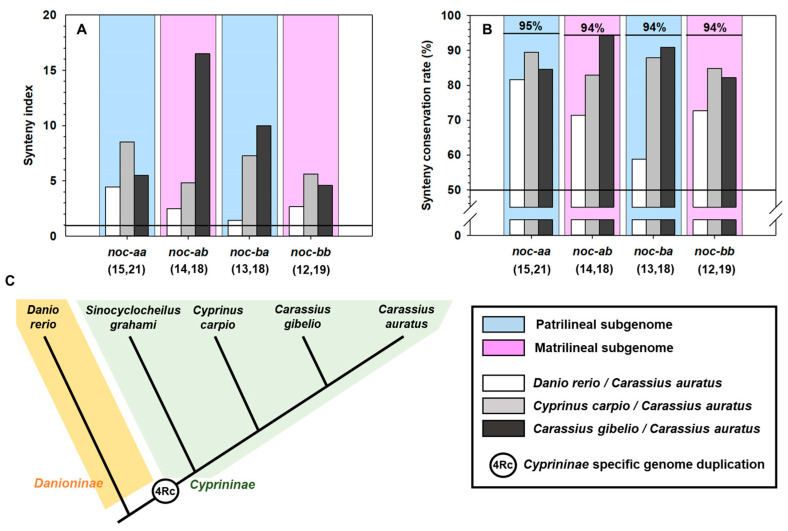
(**A**,**B**) Synteny index and synteny conservation rate in *Carassius auratus* compared to *Danio rerio* (white), *Cyprinus carpio* (light gray) and *Carassius gibelio* (dark gray). Numbers in parentheses indicate genes upstream and downstream of the *noc* paralogue analyzed. Blue and pink backgrounds indicate patrilineal or matrilineal origin of chromosomes, respectively. The lower horizontal line indicates the threshold (1 for synteny index, 50% for synteny conservation rate), where the conserved loci positions are more frequent than the non-conserved. The line above in the graphs in B indicates the maximum synteny conservation rate (%) for each paralogue. (**C**) Assumed phylogeny of species analyzed in the synteny study.

**Figure 12 ijms-25-00054-f012:**
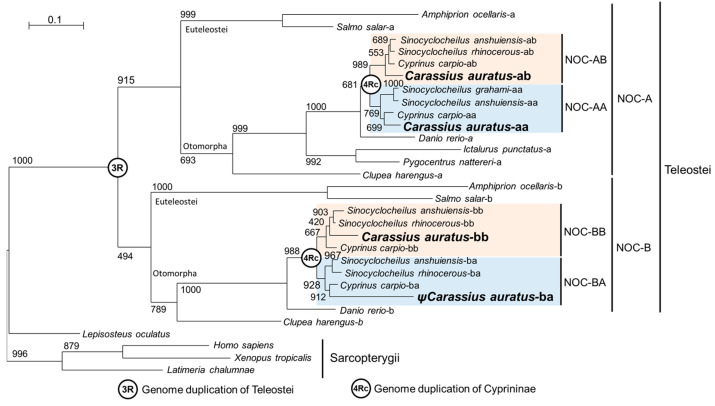
Phylogenetic tree showing the 4Rc relationships among NOC sequences. The evolutionary model used was the Jones–Taylor–Thornton, Gamma-distributed (JTT + G). The tree was inferred by the maximun likelihood method (ML). The numbers in the nodes refer to bootstrap values of a total of 1000 replicates. The scale bar indicates the average number of substitutions per position. The binomial name of the species is given on the right side of the tree. Letters A and B indicate NOC isoforms in teleosts. 3R and 4Rc indicate the proposed whole-genome duplication events in Teleostei and Cyprininae, respectively. Blue and pink boxes indicate the patrilineal of matrilineal clades of Cyprininae species, respectively. Ψ, indicates pseudogene. Species names and GenBank accession numbers of the sequences are indicated in Appendix A.

**Figure 13 ijms-25-00054-f013:**
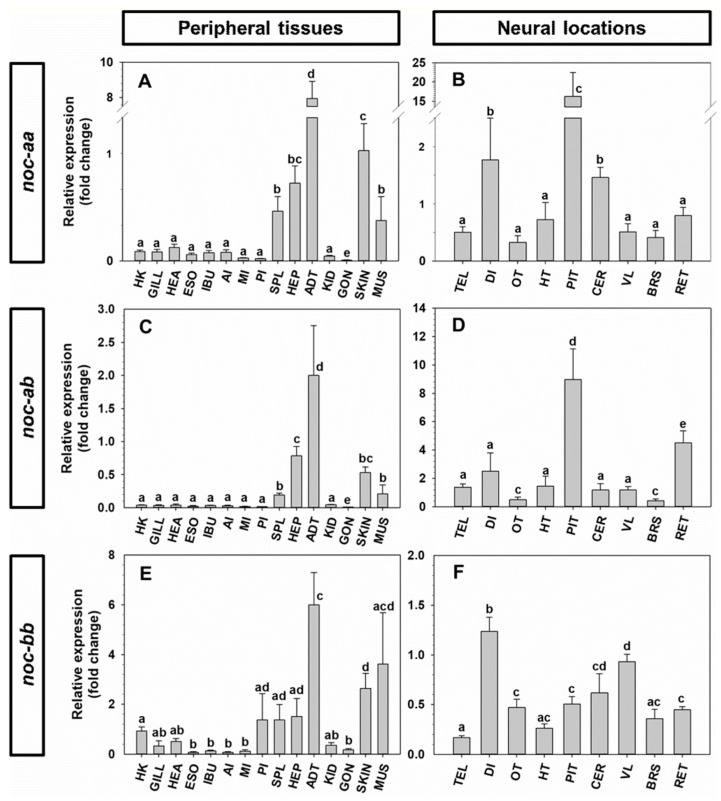
Expression of *noc-aa, noc-ab* and *noc-bb* in peripheral (**A**,**C**,**E**) and neural tissues of goldfish. Data are expressed as mean + S.E.M. (*n* = 6) relative to the hepatopancreas (**A**,**C**,**E**) and to the diencephalon (**B**,**D**,**F**). Different letters indicate statistical differences among tissues (ANOVA-SNK; *p* < 0.05). Head kidney (HK), gill (GILL), heart (HEA), esophagus (ESO), intestinal bulb (IBU), anterior intestine (AI), middle intestine (MI), posterior intestine (PI), spleen (SPL), hepatopancreas (HEP), adipose tissue (ADT), caudal kidney (KID), gonad (GON), skin (SKIN), muscle (MUS), telencephalon (TEL), diencephalon (DI), optic tectum (OT), hypothalamus (HT), pituitary (PIT), cerebellum (CER), vagal lobe (VL), brainstem (BRS), retina (RET).

**Figure 14 ijms-25-00054-f014:**
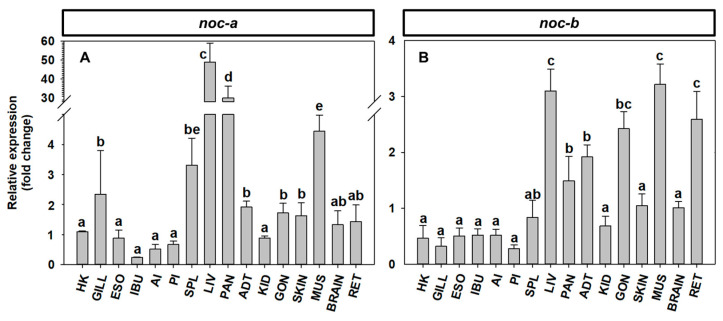
Expression of *noc-a* (**A**) and *noc-b* (**B**) in peripheral tissues and brain in zebrafish. Data are expressed as mean + S.E.M. (*n* = 5) relative to the brain. Different letters indicate statistical differences among tissues (ANOVA-SNK; *p* < 0.05). Head kidney (HK), gill (GILL), esophagus (ESO), intestinal bulb (IBU), anterior intestine (AI), posterior intestine (PI), spleen (SPL), liver (LIV), pancreas (PAN), adipose tissue (ADT), caudal kidney (KID), gonad (GON), skin (SKIN), muscle (MUS), brain (BRAIN), retina (RET).

## Data Availability

Data are included in the article and the Appendix A.

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
