# Peer review of "Gene Characterization of Nocturnin Paralogues in Goldfish: Full Coding Sequences, Structure, Phylogeny and Tissue Expression"

_ijms, 2023, doi:10.3390/ijms25010054_

Round 1
Reviewer 1 Report
Comments and Suggestions for Authors
This manuscript by Madera et al. addresses a quite narrow issue, the detailed evolution of noc genes in goldfish and allies, but it does it in a very comprehensive way. Moreover, the reporting is pretty good (text, figures, tables), to the exception of a number of minor English issues. Finally, I liked the new indices for quantifying synteny conservation even though better reference to the corresponding literature should be added.
I only have one major comment, as a phylogeneticist. The tree should really be recomputed using maximum likelihood, either on the amino-acid sequences or, even better, on nucleotide sequences, but only using the first and second positions of the codons. You can use PhyML, RAxML or IQ-TREE, for example, I don't care. Let the software choose the best evolutionary model or use a dedicated software such as modeltest. In any case, my hope would be an even better topology of the tree and a better positioning of the 3R event. Currently noc-b sequences are paraphyletic due to Euteleostei noc-b genes being attracted to noc-a sequences. Also, it could be useful to include the goldfish ba pseudogene, especially in a ML tree. The software should manage. Finally, the alignment(s) should be provided as supplementary files.
All my other comments are minor. There are indicated in the annotated version of the manuscript according to the following coloring scheme: yellow highlighting is for comments (scientific inquiries, required clarifications or suggestions of better wording: you have to read them), underlined green is for English issues (incorrect tense, missing word, inadequate use of singular or plural etc) and strike-through red is for extraneous words or sentence parts (that should be deleted).
Good work. I wish you good luck for revising the manuscript.

Comments on the Quality of English Languagesee above
Author Response
Response to Reviewer 1:
Comment 1: This manuscript by Madera et al. addresses a quite narrow issue, the detailed evolution of noc genes in goldfish and allies, but it does it in a very comprehensive way. Moreover, the reporting is pretty good (text, figures, tables), to the exception of a number of minor English issues. Finally, I liked the new indices for quantifying synteny conservation even though better reference to the corresponding literature should be added.
Response 1: Thank you very much for the kind words you shared in your review, and for taking the time to review the manuscript.
After reviewing the published methodology on synteny quantification, we found that the main goal of these bioinformatics tools is to analyze whole genomes and find conserved "synteny blocks" in two or more species. New references of other methods for quantifying synteny are added and commented in manuscript [41, 42, 43]. Our goals are simpler and more direct. From synteny maps of chromosomal segments of moderate length (only 10-30 genes) containing our gene of interest, we want to implement a simple hand-counted method (no computer-aided calculations needed) but sensitive enough to help us identify and discriminate between regions with high homology, and to help us correctly assign the actual orthology to paralogues of the same gene originated in a recent duplication.
Comment 2. I only have one major comment, as a phylogeneticist. The tree should really be recomputed using maximum likelihood, either on the amino-acid sequences or, even better, on nucleotide sequences, but only using the first and second positions of the codons. You can use PhyML, RAxML or IQ-TREE, for example, I don't care. Let the software choose the best evolutionary model or use a dedicated software such as modeltest. In any case, my hope would be an even better topology of the tree and a better positioning of the 3R event. Currently noc-b sequences are paraphyletic due to Euteleostei noc-b genes being attracted to noc-a sequences. Also, it could be useful to include the goldfish ba pseudogene, especially in a ML tree. The software should manage. Finally, the alignment(s) should be provided as supplementary files.
Response 2: We already have calculated phylogenetic tree with other methods (ML, ME, MP), but we obtained similar results. We followed your advice, and we include the sequence alignment for tree construction in supplementary files (now Supplementary figure S3). We mention of the evolutionary model recommended by MEGA11, Jones-Taylor-Thornton Model, Gamma Distributed (JTT+G). The phylogenetic tree using ML shows a similar resolution of branches, and as you anticipated, the 3R event is well positioned. Now, all teleost noc-b genes are in the same cluster, although with low support (bootstrap 494). We included too the goldfish noc-ba pseudogene and it appears in a very long branch but inside the NOC-BA cluster.
Comment 3: All my other comments are minor. There are indicated in the annotated version of the manuscript according to the following coloring scheme: yellow highlighting is for comments (scientific inquiries, required clarifications or suggestions of better wording: you have to read them), underlined green is for English issues (incorrect tense, missing word, inadequate use of singular or plural etc) and strike-through red is for extraneous words or sentence parts (that should be deleted).
Response 3: All the comments have been addressed in the revised manuscript. Your feedback provided insightful comments that I truly appreciate and clearly improved the manuscript. Thank you very much. We follow the notes included in revised manuscript, and we have made the suggested modifications in de new version (highlighted in yellow). Some comments deserve a more detailed explanation:
Line 13: what about fungal outgroups?. Good question, we found in geneBank some sequences labeled as nocturnin in fungi. But sequences are so divergent that is hard to decide if those are orthologous nocturnins or just are part of the superfamily. A lot of work is needed to assess the origin of nocturnin, and right now we think the is better not talking about nocturnin beyond metazoan.
Line 159: I get the idea but this might be wrong if one of the genomes was assembled used the other one as the reference (or source of annotation); alternatively, these pseudogenes are on their way to get lost (like in goldfish). Answer: Yes, we thought in this possibility too, but in mRNA SRA libraries there are a few fragments of goldfish noc-ba with the same mutations found in genomic DNA, then we assumed that mutations are real and present in genome.
line 125: clarify the phylogenetic reasoning for readers not familiar with teleosts phylogeny. Answer: True, we are trying to use only the name of the large groups of teleosts, and use the most accepted name, although sometimes the name of the taxon does little to indicate which fish we are talking about.
Line 442 and 479: not sure to understand why the reference tissues are not scaled to 1.0. Answer: True, is just a mathematical consequence of ΔΔCt normalization. In the last step of calculation, ΔΔCt values are transformed in 2-ΔΔCt, this exponential transformation makes that if you want to obtain a mean of 1.0 in your final data, you must calculate the geometric mean, instead of arithmetic mean (which is the mean in the figures of tissue expression). Then, arithmetic mean is always higher than 1.0, a little higher when the error is small, but a lot higher than 1.0 when the error of the mean is high.
Line 546: low relatively to what? and why only SRA libraries and not also your own RT-PCR experiments? because these are not QPCR? Answer: This was a big mistake in the manuscript, thank you, we deleted it.
Line 673: this is interesting; can you get the information across up to genome annotation in public databases? Answer: The best option to detect possible annotation errors, although is not sure to 100%, is comparing genomes of related species (same species different genotype, or same genus, or even old assembly projects). For Carassius, we have now Carassius auratus, Carassius gibelio and Carassius carassius (this last one is the low quality). We can consider that the “artefacts” must be only in one genome project, and not shared with others. That works for me.
Final comment: Good work. I wish you good luck for revising the manuscript.
Responese: Thank you very much
Reviewer 2 Report
Comments and Suggestions for Authors
It is suggested that nocturin is an important functional gene in vertebrates, which is involved in regulating energy metabolism, and may be a target for treating obesity and regulating metabolism. This gene family may also play an important role in Cyprinidae fish, and genome-wide accurate identification, localization and expression analysis are the basic work. The results are detailed and convincing, and have a good reference value for researchers in this field. But there are still some issues that need to be improved to meet the publication requirements. Details are as follows:
In Introduction section
1. the introduction unrelated to the function of the gene in fish is too long, and it is recommended to compress and adjust the relevant knowledge from the perspective of the Cyprinidae fish.
In Results section
2. “2.1. Teleostean nocturnins. Evolutive origin”, too many results are given in the main text, the topic inhere is not clear, and it is recommended to put the analysis of the basic structure unrelated to the fish sequences in the supplementary material.
3. “2.2. Nocturnin sequences in goldfish”, Please improve the presentation of Figures 4 and 5 to make the differences between the two sequences as intuitive as possible. If possible, it is recommended to change the specific sequence information into the supplementary materials to enhance readability.
In Conclusions section
4. The conclusion is too complicated, so it is suggested to further condense, retain the conclusion directly supported by the result data, and put other supporting words into the Discussion section or remove them directly.
Comments on the Quality of English Language
Moderate editing of English language required
Author Response
Response to Reviewer 2:
It is suggested that nocturin is an important functional gene in vertebrates, which is involved in regulating energy metabolism, and may be a target for treating obesity and regulating metabolism. This gene family may also play an important role in Cyprinidae fish, and genome-wide accurate identification, localization and expression analysis are the basic work. The results are detailed and convincing, and have a good reference value for researchers in this field. But there are still some issues that need to be improved to meet the publication requirements. Details are as follows:
Answer: Thank you very much for taking the time to review the article. Your feedback provides insightful comments that we truly appreciate and clearly improved the manuscript. Changes in the revised manuscript are highlighted in yellow.
In Introduction section
Comment 1: the introduction unrelated to the function of the gene in fish is too long, and it is recommended to compress and adjust the relevant knowledge from the perspective of the Cyprinidae fish.
Response 1: We attempt to summarize the current and scarce knowledge about nocturnin other than in mammals. Fish are an abandoned group in this matter, and we simply introduce the possible functions of nocturnins that can develop this family of enzymes in fish, comparing them with what has been described in other groups. Cyprinids have been our animal model for many years, and that's where we started our study.
In Results section
Comment 2: 2.1. Teleostean nocturnins. Evolutive origin”, too many results are given in the main text, the topic inhere is not clear, and it is recommended to put the analysis of the basic structure unrelated to the fish sequences in the supplementary material.
Response 2: Starting from the fact that goldfish is polyploid, we already expected a complex situation with multiple members of the nocturnin family. To address this problem, we need to explore what the situation was of what we might find, comparing different species of fish and other vertebrates using the information available in the different genome projects. We have included in the main text the key information needed to understand the rest of the manuscript, and the information we consider ancillary is in supplementary material. Other reviewers take the opposite view and recommend putting some supplementary material in the main text.
Comment 3: “2.2. Nocturnin sequences in goldfish”, Please improve the presentation of Figures 4 and 5 to make the differences between the two sequences as intuitive as possible. If possible, it is recommended to change the specific sequence information into the supplementary materials to enhance readability.
Response 3: To clarify differences between the two sequences of the splicing variants, we have modified the names of exons from spotted gar nocturnin (Fig.1). Instead of exon 1 we used exon 1a, exon 1b, and exon 1c for the three alternatives for the first exon in mature mRNA. These changes have been introduced throughout the revised manuscript. The presentation of sequences (fig. 4 fig. 5) has been modified to clearly separate two exons (exon1a and exon1b) involved in the alternative splicing mechanisms.
In Conclusions section
Comment 4. The conclusion is too complicated, so it is suggested to further condense, retain the conclusion directly supported by the result data, and put other supporting words into the Discussion section or remove them directly.
Response 4: For conclusions, we follow the journal instructions:
“This section is not mandatory but can be added to the manuscript if the discussion is unusually long or complex”.
We tried to resume the long discussion in a few final ideas. We hope that it were clear enough..
Comment 5: Moderate editing of English language required
Response 5: We have made all the suggested modifications in de new version of manuscript (highlighted in yellow)
Reviewer 3 Report
Comments and Suggestions for Authors
Review of “Gene characterization of nocturnin paralogues in goldfish: full coding sequences, structure, phylogeny and tissue expression.” By Diego Madera et al 2023
Authors present a bioinformatic characterization of nocturin paralogues in goldfish and analyzed their pattern of expression in the different tissues. However, author did not mention possible impact of circadian cycle, that would be interesting to analyze. However, the genomic context and characterization of paralogues provide a very interesting view for the evolution of this gene family, The phylogenetic analysis was done at a very basic level, the confirmation for this tree by different statistical approaches would not limit the phylogenetic analysis, Below follow some commentaries regarding the manuscript:
Methods:
Authors should refer which of the zebrafish (AB? TL?...), and Goldfish genotypes were used for the experiments.
Authors described in methods that the phylogenetic tree was obtained NJ, which is a phylogenetically weaker statistical approach for a phylogenetic analysis. Results would be much more robust if the same phylogenetic tree is obtained from 2 or 3 independent statistical approaches, adding to NJ for example Maximum Likelihood (ML) and Bayesian Inference. In addition, authors did not mention which statistical distribution was used in NJ or what was the substitution model and rates and pattern that was used… It would be necessary to include more information about construction of phylogenetic tree and validate this tree through other statical methods for phylogeny tree constructions. Also, in the case of bootstrap values they should be expressed in percentage.
Authors mentioned “Primers for reference genes of goldfish [44] and for zebrafish nocturnins were previously described” The genes used as reference should be mentioned in the text of methods not only in the Supplementary Table S6…
Results:
Authors described “we found that the only noc gene has 3 exons and 2 introns (Gene ID: 102682701).” However, the diagram provided in figure 1 and information provided by Gene ID 102682701 showed that the gene is organized in 5 exons and 4 introns but attending to the available information for transcripts identified the first 3 exons are not transcribed at the same time. This needs to be corrected. However, I think nomenclature given to each of the exons is appropriated.
In figure 9, part of the text of the results description is between the legend and the figure. Please correct.
The analysis for the pattern of expression in the different tissues its interesting, but it would be advantageous to evaluate the expression with the context of the circadian cycle. Authors can at least discuss what is expected to happen within this context. In addition, was the circadian cycle considered regarding tissue collection from fishes? Could that have an impact on the pattern of expression that authors have described?
Author Response
Response to Reviewer 3:
Authors present a bioinformatic characterization of nocturin paralogues in goldfish and analyzed their pattern of expression in the different tissues. However, author did not mention possible impact of circadian cycle, that would be interesting to analyze. However, the genomic context and characterization of paralogues provide a very interesting view for the evolution of this gene family, The phylogenetic analysis was done at a very basic level, the confirmation for this tree by different statistical approaches would not limit the phylogenetic analysis, Below follow some commentaries regarding the manuscript:
AUTHORS: Thank you very much for taking the time to review the article. Your feedback provides insightful comments that we truly appreciate and clearly improved the manuscript. Changes in the revised manuscript are highlighted in yellow)
Methods:
Comment 1: Authors should refer which of the zebrafish (AB? TL?...), and Goldfish genotypes were used for the experiments.
Response 1: This information is now included in “4.1. Animal maintenance”:
Line 773: Goldfish (comet goldfish, Carassius auratus auratus) and zebrafish (Danio rerio, genotype Tübingen)
Comment 2: Authors described in methods that the phylogenetic tree was obtained NJ, which is a phylogenetically weaker statistical approach for a phylogenetic analysis. Results would be much more robust if the same phylogenetic tree is obtained from 2 or 3 independent statistical approaches, adding to NJ for example Maximum Likelihood (ML) and Bayesian Inference. In addition, authors did not mention which statistical distribution was used in NJ or what was the substitution model and rates and pattern that was used… It would be necessary to include more information about construction of phylogenetic tree and validate this tree through other statical methods for phylogeny tree constructions. Also, in the case of bootstrap values they should be expressed in percentage.
Response 2: We already have calculated phylogenetic tree with other methods previously (ML, ME, MP), but we obtained similar results. We followed your advice, we used the evolutionary model recommended by MEGA11, Jones-Taylor-Thornton Model, Gamma Distributed (JTT+G). The phylogenetic tree using ML shows a similar resolution of branches, and the 3R event is well positioned. Now, all teleost noc-b genes are in the same cluster, although with low support (bootstrap 494). We included too the goldfish noc-ba pseudogene, and it appears in a very long branch but inside the NOC-BA cluster. We include the sequence alignment (Clustal) for tree construction in supplementary files (now Supplementary figure S3).
For the bootstrap values we chose to put the real value (from 1000 replicates) instead of the percentage, since that is how it appears in many works, and also because from the numerical point of view it is equivalent (999 vs. 99.9).
Comment 3: Authors mentioned “Primers for reference genes of goldfish [44] and for zebrafish nocturnins were previously described” The genes used as reference should be mentioned in the text of methods not only in the Supplementary Table S6…
Response 3: This information about reference genes is now in section “4.6. Tissue expression of noc mRNA using RT-qPCR” a bibliographic reference is corrected too.
Line 915: Primers for reference genes of goldfish (β-actin and ef-1α) [47] and for zebrafish nocturnins and reference gene (β-actin) were previously described [5, 48].
Results:
Comment 4: Authors described “we found that the only noc gene has 3 exons and 2 introns (Gene ID: 102682701).” However, the diagram provided in figure 1 and information provided by Gene ID 102682701 showed that the gene is organized in 5 exons and 4 introns but attending to the available information for transcripts identified the first 3 exons are not transcribed at the same time. This needs to be corrected. However, I think nomenclature given to each of the exons is appropriated.
Response 4: We have modified the names of exons from spotted gar nocturnin (Fig.1). Instead of exon 1 we used exon 1a, exon 1b, and exon 1c for the three alternative sequences for the first exon in mature mRNA (after the alternative splicing). These changes have been included in the rest of the manuscript.
Comment 5: In figure 9, part of the text of the results description is between the legend and the figure. Please correct.
Response 5: Small changes in the text may modified the text line number and moved the figures. The transformation of manuscript from Word to PDF format may move the lines too. We careful checked the position of figures inside the main text. We hope that these editing issues will be resolved in the final version.
Comment 6: The analysis for the pattern of expression in the different tissues its interesting, but it would be advantageous to evaluate the expression with the context of the circadian cycle. Authors can at least discuss what is expected to happen within this context. In addition, was the circadian cycle considered regarding tissue collection from fishes? Could that have an impact on the pattern of expression that authors have described?
Response 6: We agree with the reviewer on the interest of studying the circadian aspects of nocturnin expression. We are conscious of the main importance of circadian regulation on nocturnin expression, in fact this is the main objective for the next work we are trying to publish.
However, taking in mind the existence of multiple paralogs of nocturnin in Cyprininae, it is important firstly to characterize such paralogs before to investigate circadian changes in its expression. We can advance that the circadian rhythm regulation is complex in goldfish were there are arhythmic tissues, but others are rhythmic with low or high amplitude.
In this manuscript sampling was performed at zeitgeber time 4 (ZT4) just for identifying the most promising tissues with a significant expression of all nocturnin paralogues in goldfish. This information has included in the revised manuscript (lines 905 and 909)
Round 2
Reviewer 2 Report
Comments and Suggestions for Authors
I acknowledge the author's efforts and feel that the quality of the manuscript has greatly improved and meets the conditions for publication. It is recommended that the editorial department accept the current version of the manuscript.